# A glomerulus-on-a-chip to recapitulate the human glomerular filtration barrier

Astgik Petrosyan[1], Paolo Cravedi[2], Valentina Villani[1], Andrea Angeletti [3], Joaquin Manrique[4], Alessandra Renieri [5], Roger E. De Filippo[1,6], Laura Perin[1,6,7] & Stefano Da Sacco [1,6,7]

In this work we model the glomerular filtration barrier, the structure responsible for filtering the blood and preventing the loss of proteins, using human podocytes and glomerular endothelial cells seeded into microfluidic chips. In long-term cultures, cells maintain their morphology, form capillary-like structures and express slit diaphragm proteins. This system recapitulates functions and structure of the glomerulus, including permselectivity. When exposed to sera from patients with anti-podocyte autoantibodies, the chips show albuminuria proportional to patients' proteinuria, phenomenon not observed with sera from healthy controls or individuals with primary podocyte defects. We also show its applicability for renal disease modeling and drug testing. A total of 2000 independent chips were analyzed, supporting high reproducibility and validation of the system for high-throughput screening of therapeutic compounds. The study of the patho-physiology of the glomerulus and identification of therapeutic targets are also feasible using this chip.

[1] GOFARR Laboratory for Organ Regenerative Research and Cell Therapeutics in Urology, Saban Research Institute, Division of Urology, Children's Hospital Los Angeles, Los Angeles, CA, USA. [2] Division of Nephrology, Icahn School of Medicine at Mount Sinai, New York, NY, USA. [3] Department of Experimental, Diagnostic and Specialty Medicine, Nephrology, Dialysis and Renal Transplant Unit, S. Orsola-Malpighi Hospital, University of Bologna, Bologna, Italy. [4] Nephrology Service, Complejo Hospitalario de Navarra, Pamplona, Spain. [5] Genetica Medica, Azienda Ospedaliera Universitaria Senese, Siena, Italy. [6] Department of Urology, Keck School of Medicine, University of Southern California, Los Angeles, CA, USA. [7] These authors contributed equally: Laura Perin, Stefano Da Sacco. Correspondence and requests for materials should be addressed to L.P. (email: lperin@chla.usc.edu) or to S.D.S. (email: sdasacco@chla.usc.edu)

Over 10% of adults worldwide are affected by renal abnormalities and the number of those with end-stage renal disease (ESRD) receiving replacement therapy with dialysis or transplant is estimated at >1.4 million, with an annual growth rate of 8%[1]. Major progresses in understanding environmental and genetic risk factors as well as pathogenic mechanisms of renal disease progression have been accomplished, but outcomes of affected individuals have not appreciably improved over the last two decades[1].

Despite a wide variety of causes including metabolic abnormalities, hypertension, autoimmunity, and genetic background, a common early pathologic hallmark of chronic kidney disease (CKD) is decreased glomerular filtration and loss of functional glomeruli[2]. The main function of the glomeruli is to filter fluids and electrolytes from the blood, while retaining plasma proteins[3]. This activity happens at the level of the glomerular filtration barrier (GFB) and is coordinated by the interaction of two highly specialized glomerular cells (the fenestrated endothelium and the podocytes), which are separated by a thin layer of glomerular basement membrane (GBM[4]).

One of the major roadblocks to the development of successful therapeutics for CKD depends on the ability to effectively establish 3D models that can mimic the complex structure and function of the GFB. Even though some success in generating kidney structures has been described using conventional 2D or 3D culture systems (including spheroids and extracellular based gels[5]), the results are still inconsistent. The recent discovery of kidney organoids allows the formation of nephron-like structures that recapitulate some of the characteristics of the glomerular environment[6], but they have no or limited filtration activity and the deposition of a correct GBM has not been fully demonstrated yet. Most importantly, the cells used to generate these organoids are derived from genetically modified cell lines and require complex nephrogenic induction protocols, which may affect their morphology and function[7].

Recently, the development of microfluidic platforms (organ on a chip) that allow co-culture of cells and matrices, combined with the application of perfusion and spatial control over signaling gradients[8], have been used for physiological studies and drug discovery for many complex organs including liver, heart, gut, lung, and brain[9–15]. The chip technology has been used to replicate renal structures, including proximal tubules[16–19] and, in few instances, the glomerular compartment[20–22]. However, in the majority of the current glomerular chips, podocytes and glomerular endothelial cells are separated by a synthetic membrane usually constituted by polydimethylsiloxane[8,23,24]. While these membranes are equipped with openings (pores) that allow free exchange of media and growth factors, they do not allow the proper crosstalk between glomerular cells that is key for GFB function.

Herein, we describe a glomerulus-on-a-chip (referred to as GOAC) constituted by human podocytes and human glomerular endothelial cells (hGEC) seeded on Organoplates[TM] (MIMETAS). Our system is characterized by the absence of an artificial membrane separating the two monolayers. Cells can be cultured in these chips for long term, maintaining their phenotype, and glomerular cells can properly interact to generate layer of extracellular matrix composed by collagen IV trimer and laminin, the major constituents of the GBM in vivo. Such GFB-like structure recapitulates function of the GFB, including selective permeability and response to nephrotoxic compounds. We validated specific functionality of these chips using serum from individuals affected by different glomerular diseases, including membranous nephropathy (MN) and evaluated drug response.

We also assessed response of GOAC to glucose-induced damage and performed studies of disease modeling by

generating GOAC using amniotic fluid kidney progenitor-derived podocytes (hAKPC-P[25]) from subjects affected by Alport syndrome (AS), a hereditary CKD characterized by mutations in the alpha chains of COL4 genes[26]. Chips generated using these AS podocytes present impaired permselectivity to albumin, due to a dysfunctional assembly of the GBM, typical of AS.

## Results

**Characterization of human podocytes and hGEC.** In this work, we used different types of podocytes of human origin: (1) primary podocytes (hpPOD); obtained from discarded kidneys harvested from patients with non-nephrological cause of death, thus the cells were healthy; (2) immortalized podocytes (hiPOD) considered for many years the gold standard for in vitro cultures[27,28]; and (3) amniotic fluid-derived podocytes (hAKPC-P): obtained in our laboratory as published[25]. hAKPC-P can be derived with minimal cell manipulation and, before differentiation, can be expanded for many passages while maintaining their ability to differentiate into podocytes with high efficiency.

hpPOD were obtained from human glomeruli and positively selected for nephrin (Supplementary Figs. 1a, b and 2a) and were seeded immediately after isolation or after one passage in culture. hAKPC-P (Supplementary Fig. 2b,) and hiPOD were differentiated in VRADD media on collagen I prior to seeding on the chip[28]. Podocyte morphology is evident in all three lines (Supplementary Fig. 2c–e) as well as expression of markers typical of mature podocytes such as WT1 and the slit diaphragm protein nephrin (Supplementary Fig. 2f–k) (Fig. 1a–f), while they were negative for CD31 (endothelial marker) (Fig. 1g–i; Supplementary Fig. 2l–n) and wheat germ agglutinin (WGA, identifying the endothelium glycocalyx), overall confirming their podocyte phenotype (Supplementary Fig. 2o–q).

The glomerular endothelium is characterized by unique fenestrations that can be considered an analogous of podocyte filtration slits and contributes to the GFB function. Primary hGEC, isolated from the same kidneys from which hpPOD were derived, were negative for podocyte markers (WT1, nephrin) and positive for CD31 and vascular endothelial growth factor receptor 2 (VEGFR2; this receptor is expressed in vivo by GEC since they highly respond to the VEGF gradient signaling from podocytes[4]) and WGA (Fig. 1j–l; Supplementary Fig. 3a, b, e–i). hGEC were also found to be positive for EH domain containing 3 (EHD3, Supplementary Fig. 4a, b), a marker specifically expressed by the human glomerular endothelium in the kidney (Supplementary Fig. 4c[29]). These hGEC are characterized by the presence of fenestrations (with an average diameter of 60.55 nm ± 3.35 SEM, compatible with measurements performed in previous studies[30,31] (Supplementary Fig. 4d). Positive expression for major glycocalyx components like Syndecan-1 (Supplementary Fig. 4e, f), Syndecan-4 (Supplementary Fig. 4g, h), and heparan sulfate (Supplementary Fig. 4i) was also assessed[32]. As negative controls for podocytes and hGEC, we used human lines of fibroblasts (hFIB) and human lung endothelial cells (HuLECs), respectively. Both HuLECs and hFIB were negative for WT1 and nephrin (Fig. 1m, n, p, q; Supplementary Fig. 3j, k; o, p); HuLEC were positive for CD31, VEGFR2, and WGA (Fig. 1o; Supplementary Fig. 3l–n) while hFIB were negative for all these markers (Fig. 1r; Supplementary Fig. 3q–s).

**Culturing human podocytes and hGEC on the chip.** We first investigated whether our system supports the culture of hGEC and podocytes separately. A schematic representation of the chip and channel seeding is shown in Fig. 1s, t. Since collagen I stratification present in channel E is achieved by meniscus pinning, there is no artificial membrane between the perfusion lane

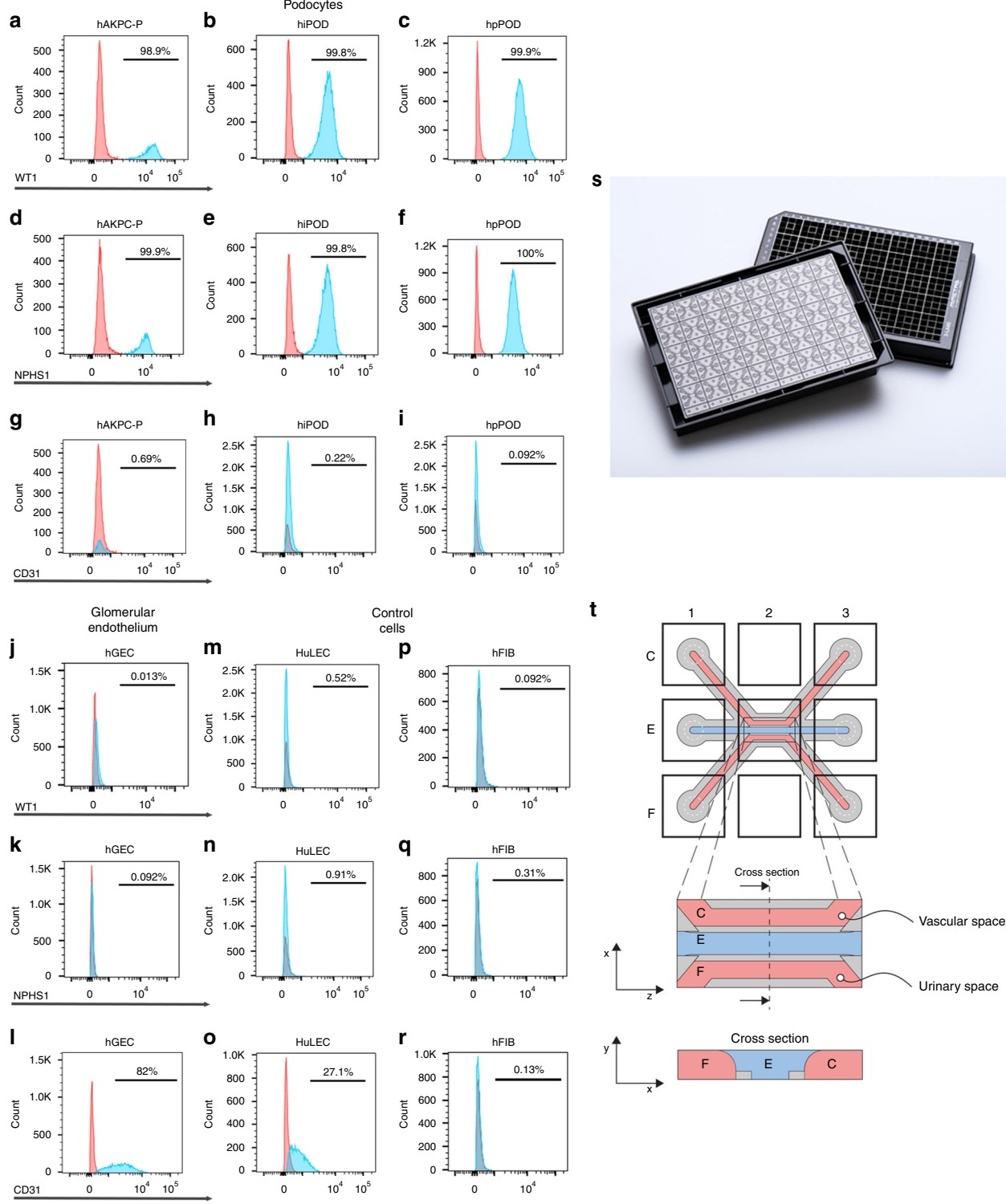

and the collagen. Therefore, the interaction of the layers of seeded cells (channel C) and matrix recapitulates the in vivo GFB oriented from endothelial cells, the GBM, podocytes, and the urinary space of Bowman's capsule (channel F). hGEC were seeded in channel C (Fig. 2a) and cultured in endothelial medium. We confirmed their ability to form a capillary-like structure in the chip (Fig. 2b) and maintain expression of endothelial marker CD31 (Fig. 2c). Presence of an endothelial glycocalyx on the surface was also confirmed by immunofluorescent staining using WGA (Fig. 2d). Thickness of the glycocalyx was confirmed to be ~0.5 µm (Supplementary Fig. 4j), compatible with results previously reported by other groups on human immortalized glomerular endothelial cells[33] and in vivo[32].

hAKPC-P, hiPOD, or hpPOD were seeded in channel C (Fig. 2e) and cultured in VRADD media. Confocal imaging revealed that hAKPC-P, hpPOD, and hiPOD expressed nephrin prevalently in primary processes (Fig. 2f–h, magenta arrows), which appeared less organized in hiPOD (Fig. 2g) compared to

**Fig. 1** Characterization of cellular lines and description of the microfluidic chip. **a–i** Flow cytometry for WT1 (PE, red) in hAKPC-P (**a**), hiPOD (**b**), and hpPOD (**c**); for nephrin (NPHS1-FITC, green) in hAKPC-P (**d**), hiPOD (**e**), and hpPOD (**f**) and for CD31 (FITC, green) in hAKPC-P (**g**), hiPOD (**h**), and hpPOD (**i**). All three lines show almost 100% expression of podocyte markers and absence of endothelial marker confirming their podocyte phenotype. **j–l** Flow cytometry for WT1 (PE, red; **j**), for nephrin (NPHS1-FITC, green; **k**), and for CD31 (FITC, green; **l**) in hGEC show low expression (<1%) of podocyte markers and higher expression of endothelial marker confirming their endothelial phenotype. **m–r** Flow cytometry for WT1 (PE, red) in HuLEC (**m**) and hFIB (**p**), for nephrin (NPHS1-FITC, green) in HuLEC (**n**) and hFIB (**q**), and for CD31 (FITC, green) in HuLEC (**o**) and hFIB (**r**). HuLEC and hFIB are both negative for podocyte makers, HuLEC are positive for endothelial marker while hFIB are negative. Unstained control is shown in red while stained sample is shown in light blue. **s, t** The chip is a microfluidic layer sandwiched between two 175 μm glass plates (OrganoPlate™ platform, courtesy of MIMETAS™, panel **s**). The three-channel version of the Organoplate™ comprises 40 networks on one 96-well format plate. **t** The central section is subdivided into three channels by a system known as PhaseGuide™, a thin 30 μm tall ridge that acts as a pinning barrier for incoming fluids[23]. Following the filling of channel E with collagen I, channel C (representing the vascular space) is seeded with cells and then filled with growth medium; the channel F (representing the urinary space) is where the filtrate is collected. Cross-section depicts patterning of cells and collagen within the GOAC. Flow of culture medium is achieved by leveling between the media reservoirs of the lanes C and F. The platform is placed on an interval rocker, with an angle to assure leveling. By changing the angle of the platform (rocker settings: interval = 8 min, angle = 7°), the direction of fluid flow is reversed

hAKPC-P (Fig. 2f) and hpPOD (Fig. 2h). Taken together, these results indicate that hGEC and podocytes can be cultured in the chip maintaining their morphology and phenotype.

**Structural characterization of GOAC**. We next co-cultured podocytes and hGEC to generate the GOAC (Fig. 2i). First, we filled channel E with collagen I and, after gelification, we seeded podocytes in channel C. Within 20 min, they started layering on the side of the collagen wall. After 24 h, all cells firmly attached to the wall to form a monolayer to cover the collagen surface. The addition of hGEC was performed on the top inlet in channel C. After 24 h, the chip was placed under flow conditions and hGEC started forming a continuous capillary-like layer that is evident as soon as day 5 in co-culture (Supplementary Fig. 5a–i). When CM-DiI-labeled podocytes (green) and CFSE-labeled hGEC (magenta) were seeded together, they showed the ability to form clearly distinguishable layers (Supplementary Fig. 5j). Importantly, selective expression of nephrin and CD31, respectively, was confirmed by confocal microscopy (Fig. 2j–l). Cells can be co-cultured for at least 4 weeks, maintaining their viability (Supplementary Fig. 5k–m), thus confirming that our seeding strategy allows long-term maintenance of the cell phenotype in the chip. Following successful filling with collagen I our overall successful rate for establishing the chip evaluated by visual observation was 81% (hAKPC-P + hGEC: 81.9% ± 3.7; hiPOD + hGEC: 88.9% ± 7.2; hpPOD + hGEC: 78.8% ± 11.1, error expressed as SEM).

Podocytes and endothelial cells alone do not guarantee the correct function of the filtration barrier in the absence of a GBM. The human GBM is characterized by the presence of collagen IV trimers, COL4α3α4α5 and in lower quantity of COL4α1α1α2 (ref. [34]), and laminins (like LAM5α2β1γ)[35]. Patients affected by mutations of these membranous proteins (like AS or Pearson Syndrome[36]) present progressive CKD. Both podocytes and GEC are necessary for the proper assembly of the GBM[35]. Podocytes are exclusively responsible for the production of COL4α3α4α5 while COL4α1α1α2 and LAM5α2β1γ are produced by both podocytes and hGEC[37]. We confirmed production and deposition of α1, α2, and α4 chains of the COL4 as well as α5 chain of LAM for both hAKPC-P + hGEC and hpPOD + hGEC chips (Fig. 2m); the hiPOD + hGEC chip did show lower expression of these proteins as confirmed by immunofluorescence (Fig. 2n–v). We further confirmed de novo deposition of GBM components collagen IV (COL4α3) and LAMA5 (Fig. 2w–y) by western blotting, thus demonstrating that our chips resemble the in vivo GFB.

In vivo, glomerular cells are subject to the mechanical stress (shear stress) generated by the blood flowing on the apical surface

of the endothelium and by the filtrate flowing from the vascular lumen to the Bowman's space[38]. Shear stress affects phenotype, behavior, and permeability of both podocytes and endothelial cells and therefore plays a key role in glomerular hemodynamics[38]. In glomerular capillaries, shear stress has been estimated to range from approximately 1 to about 95 dyn/cm$^2$ (corresponding to 0.1–9.5 Pa)[39]. The shear stress within our three-channel system, calculated based on a previous work[40], is equal to 0.0117 Pa (or 0.117 dyn/cm$^2$), a value closer to the physiological parameter compared to existing glomerulus-on-a-chip systems established in other labs for which the reported shear stress ranges from 0.003 (ref. [20]) to 0.007 dyn/cm$^2$ (ref. [22]) on the top channel.

**Permselectivity as functional measure of a working GOAC**. One of the most important characteristics of the GFB is permselectivity, i.e. the capacity to filter molecules based on their size[3,41]. Albumin is the most abundant protein in human plasma and under physiological conditions is retained within the bloodstream. Leakage of albumin in the urine is considered a sign of kidney dysfunction, and its levels (albuminuria) correlate with the severity of glomerular injury in mice and humans[3]. We tested chip permselectivity by adding a physiological concentration of FITC-conjugated albumin (40 mg/ml[42]) to the media in channel C (Fig. 3a). hAKPC-P + hGEC, hiPOD + hGEC, and hpPOD + hGEC prevented albumin leakage at 5 and 60 min (Fig. 3b–d). To test the hypothesis that the permselectivity was provided by the two contiguous layers of podocytes and hGEC, the same experiment was repeated with chips generated with (1) podocytes + HuLEC (as negative control for hGEC, Fig. 3e), (2) human fibroblasts (as negative control for podocytes) + hGEC (Fig. 3f), or (3) devoid of cells (Fig. 3g). In all these conditions, FITC-conjugated albumin easily filled all three channels at both 5 and 60 min (Fig. 3e–g). Notably, the hAKPC-P + HuLECs chip retained albumin selectively more than fibroblasts, suggesting that podocytes are possibly the main players of albumin permselectivity in the chip as already hypothesized in vivo[43]. Interestingly, hiPOD chips did not exhibit a statistically significant difference compared to chips built using hFIB, further suggesting that the immortalized line is less suitable for these assays (Fig. 3h). The lower efficiency of hiPOD permselectivity could possibly stem from to the lower expression of laminin or collagen-binding components like α3, α1, and β1 integrin chains and CD151 (which associates with cell-matrix complexes like integrins), integrin chains β3, α5, αV that instead mediate fibronectin binding and are activated in progressive CKD, as shown in our previous work[25]. These alterations do not

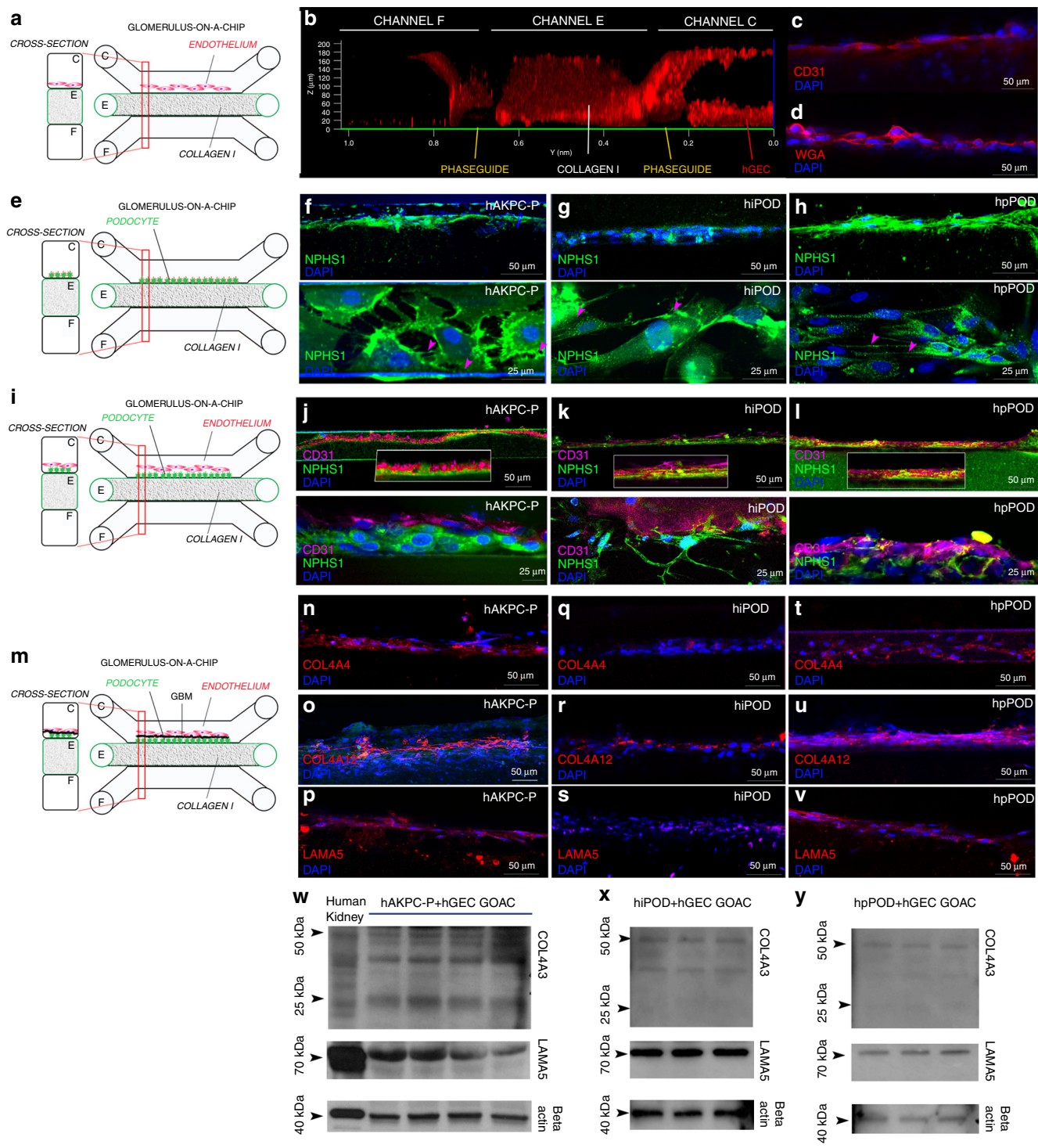

favor proper GBM production and attachment to it[44] and possibly affects their ability to assemble a strong barrier to retain albumin. Long-term analysis of GOAC permselectivity confirmed the maintenance of cell viability of the GOAC as well as efficiency of permselectivity for at least 28 days following the hGEC seeding in both hAKPC-P + hGEC and hpPOD + hGEC chips. Performance of hiPOD chips markedly decreased at 2 weeks (Fig. 3i).

To prove GOAC permselectivity, together with the capability of retaining albumin, we also tested its capacity of filtering molecules that are freely filtered by glomerulus in vivo, such as

inulin[45]. As shown in Supplementary Fig. 6a–d, the GOAC can filter inulin, thus confirming that the GOAC is constituted by a functional GFB that can accurately perform differential clearance of albumin and inulin, like the in vivo GFB.

To further support the advantages of our platform in comparison to other in vitro models and to confirm that our GOAC replicates in bona fide the semi-permeability of GFB, we have generated podocyte-endothelial barriers on 24-well transwells using the same protocol (including cell isolation, media, and timing) used for our chips. As shown in Supplementary Fig. 6e, all transwells exhibited a significant albumin leakage thus

**Fig. 2** Seeding of endothelial cells and podocytes in Organoplates™ and generation of the GOAC. **a** Representation of seeded hGEC in Organoplate™. **b** Confocal Z-stack image showing formation of capillary-like structure. Phaseguide™ components (yellow lines) can be easily identified within the panel. Channel E is filled with collagen I (visible thanks to autofluorescence in the red channel) while channel C confirms formation of a capillary-like structure by a continuous monolayer of hGEC (stained with CD31, Alexa-555, red). **c, d** confocal image for CD31 (Alexa-555, red, **c**) and for WGA (Rhodamine, red, **d**) in hGEC after 28 days of culture. **e** Representation of seeded podocytes in Organoplate™. **f–h** Confocal image for nephrin (NPHS1-FITC, green) in hAKPC-P (**f**), in hiPOD (**g**), and hpPOD (**h**) seeded on the channel C after 28 days of culture. Nephrin expression is present on the level of cell–cell contact (arrow). **i** Representation of seeded of podocytes and hGEC in Organoplate™. **j–l** Confocal image for nephrin (NPHS1-FITC, green) and CD31 (Alexa-555, magenta) in hAKPC-P + hGEC chip (**j**), in hiPOD + hGEC chip (**k**), and hpPOD + hGEC chip (**l**) after 28 days of culture. All three podocyte lines form a continuous layer, distinguishable from the hGEC layer. **m** Representation of a GBM-like structure in Organoplate™. **n–v** Confocal image for COL4A4 (Alexa-555, red), for COL4A12 (Alexa-555, red), and for LAMA5A (Alexa-555, red) in hAKPC-P + hGEC chip (**n–p**), in hiPOD + hGEC chip (**q–s**), and hpPOD + hGEC chip (**t–v**) after 28 days of culture. hAKPC-P + hGEC chip and hpPOD + hGEC chip show de novo generation of GBM, which is less evident in the hiPOD + hGEC. Nuclei are stained with DAPI (blue). All pictures: scale bar = 50 μm; except bottom panel in **f–l** with scale bar = 25 μm bar. **w–y** Western blot analysis for COL4A3 (25 kDA, monomeric form; 50 kDA, dimeric form), LAMA5 (70 kDa), and beta actin (40 kDa) in hAKPC-P + hGEC (**w**), hiPOD + hGEC (**x**), and hpPOD + hGEC (**y**) GOAC. Positive control: human whole-kidney lysate. Western blot images were cropped to show the relevant bands and improve clarity. Uncropped and unprocessed scans for western blotting analysis are provided in Supplementary Fig. 11

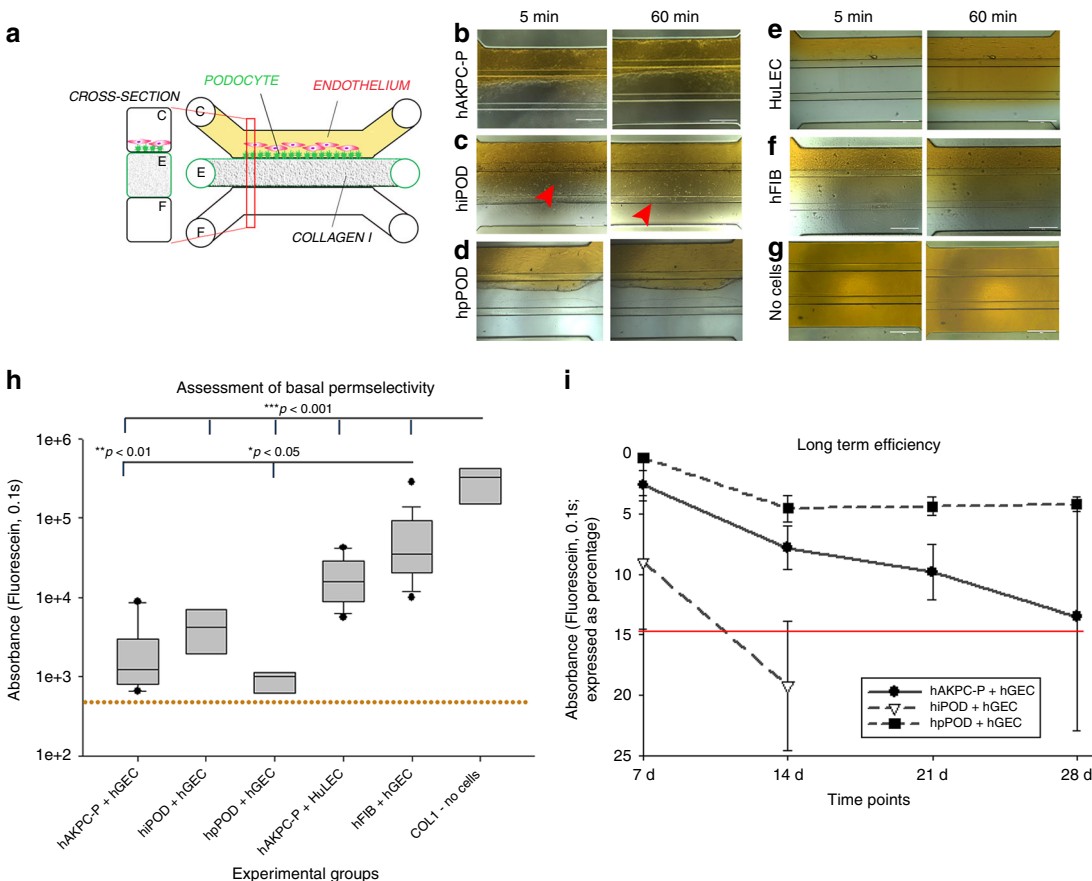

**Fig. 3** GOAC permselectivity and long-term efficiency. **a** Representation of GOAC albumin permselectivity assay: albumin-FITC (40 mg/ml) is applied to channel C (yellow) and flow-through collected in channel F. Bright field showing albumin leakage (yellow) after 5 min (left columns) and 60 min (right columns) in hAKPC-P + hGEC chip (**b**), in hiPOD + hGEC chip (**c**), in hpPOD + hGEC (**d**), in hAKPC-P + HuLEC chip (**e**); in hFIB + hGEC chip (**f**) and in chip with no cells but just collagen I in channel E (**g**). It is evident that albumin is absent only in chips generated by hAKPC-P + hGEC chip (**b**), in hiPOD + hGEC chip (**c**), in hpPOD + hGEC chip (**d**). Some leakage is present in hiPOD + hGEC chip (red arrow) and leakage is evident in chips with no cells (**g**) and in chips formed by hAKPC-P + HuLEC (**e**) and in hFIB + hGEC chip (**f**). **h** Box plot graph of fluorescein absorbance (expressed as log) in filtrate after 60 min All conditions with cells were significantly different (p < 0.001) to chips without cells. hAKPC-P + hGEC and hpPOD + hGEC chips (but not hiPOD + hGEC) were statistically significantly different (p < 0.01 and p < 0.05, respectively) from chips generated using hFIB instead of podocyte lines. Number of replicates for chips used in **h** as follow: hAKPC-P + hGEC chip: #12; hiPOD + hGEC chip: #6; hpPOD + hGEC chip: #7; hAKPC-P + HuLEC chip: #13; hFIB + hGEC chip: #19; no cell chip: #3. Significant differences were determined by a one-way ANOVA and Holm–Sidak post hoc test, *p < 0.05, **p < 0.01, ***p < 0.001. Box plots show the median, the 25th, and 75th percentiles, whiskers (median ± 1.5 times interquartile range), and outliers (solid circle). **i** Graph of fluorescein absorbance in filtrate after 60 min at 7, 14, 21, and 28 days. hAKPC-P + hGEC and hpPOD + hGEC chips maintained permselectivity efficiency at 28 days; hiPOD + hGEC permselectivity was highly reduced at 2 weeks. Red line (corresponding to a 15% loss of efficiency in retaining albumin) represents the threshold chosen as lower acceptable performance by GOAC chips. Number of replicates for chips used in **i** as follow: hAKPC-P + hGEC chip: 7d#13, 14d#10, 21d#9, 28d#4; hiPOD + hGEC chip: 7d#10, 14d#11; hpPOD + hGEC chip: 7d#7, 14d#22, 21d#15, 28d#15

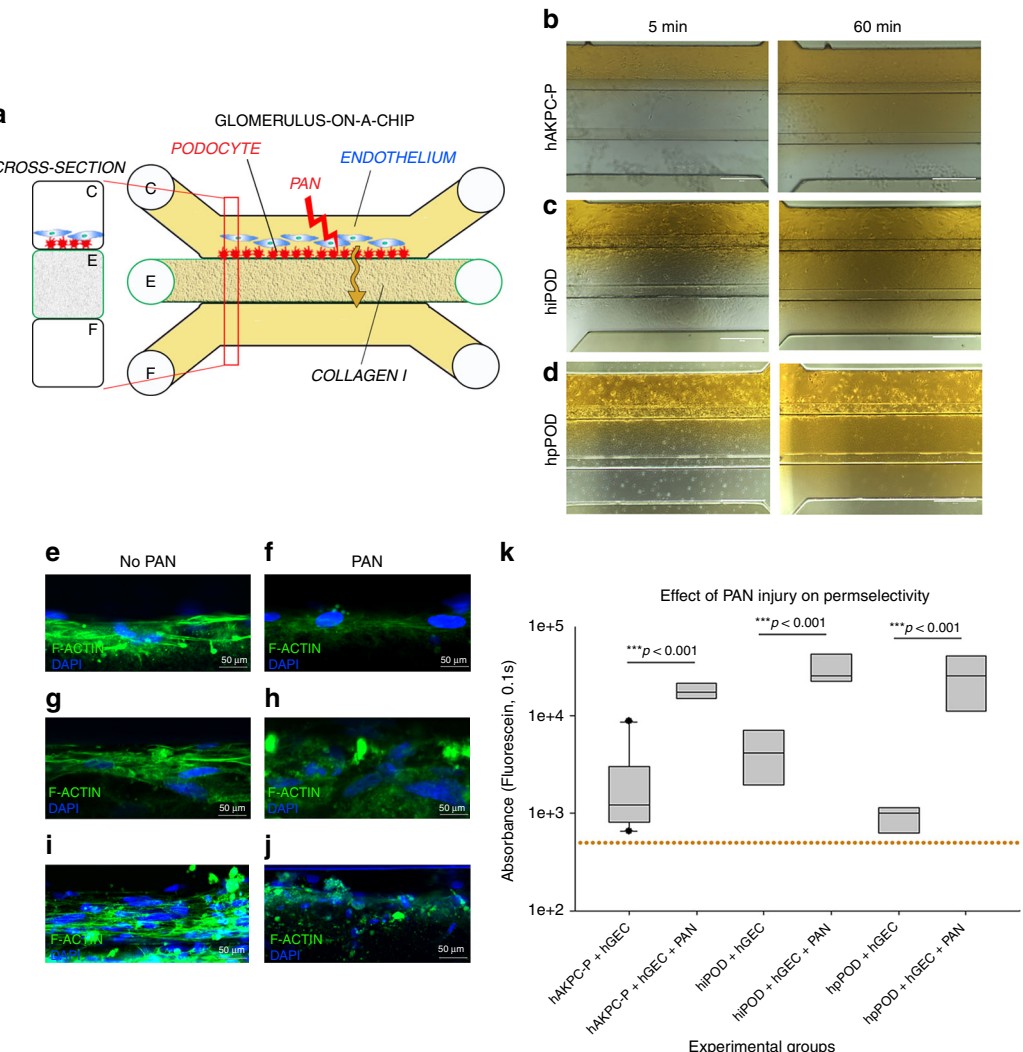

**Fig. 4** GOAC and puromycin aminonucleoside (PAN) injury. **a** Representation of GOAC albumin permselectivity assay following PAN injury: PAN (5-day induction, 10 mg/ml) is applied to channel C followed by albumin-FITC (40 mg/ml, yellow) and flow-through collected in channel F. **b–d** Bright field showing albumin leakage (yellow) after 5 min (left columns) and 60 min (right columns) in hAKPC-P + hGEC chip (**b**), in hiPOD + hGEC chip (**c**), in hpPOD + hGEC (**d**) after PAN injury. Marked albumin leakage occurs following podocyte injury. **e–j** Damage to podocytes was assessed by F-actin staining in hAKPC-P + hGEC chip before (**e**) and after PAN damage (**f**), in hiPOD + hGEC chip before (**g**) and after PAN damage (**h**), and in hpPOD + hGEC before (**i**) and after PAN damage (**j**) confirming widespread disruption of the cytoskeleton. Nuclei are stained with DAPI (blue); actin filaments stained with phalloidin (green). All pictures: scale bar = 50 μm; **k** box plot graph of fluorescein absorbance (expressed as log) in filtrate collected after 60 min after 5 days exposure to PAN. For all experimental groups, a marked and statistically significant increase was found in albumin permeability following injury. Number of replicates for chips used in **k** is as follows: hAKPC-P + hGEC chip: #12 and #4; hiPOD + hGEC chip: #6 and #4; hpPOD + hGEC chip: #7 and #3. Significant differences were determined by a one-way ANOVA and Holm–Sidak post hoc test, *$p < 0.05$, **$p < 0.01$, ***$p < 0.001$. [To improve clarity, the following significant differences were not drawn in the graph: hiPOD + hGEC + PAN vs. hAKPC-P + hGEC $p < 0.001$; hiPOD + hGEC + PAN vs. hpPOD + hGEC $p < 0.001$; hpPOD + hGEC + PAN vs. hAKPC-P + hGEC $p < 0.001$; hpPOD + hGEC + PAN vs. hiPOD + hGEC $p < 0.001$; hAKPC-P + hGEC + PAN vs. hpPOD + hGEC $p < 0.001$; hAKPC-P + hGEC + PAN vs. hiPOD + hGEC $p < 0.05$; hiPOD + hGEC + PAN vs. hAKPC-P + hGEC + PAN $p < 0.05$.] Box plots show the median, the 25th and 75th percentiles, whiskers (median ± 1.5 times interquartile range), and outliers (solid circle)

suggesting that, under the same conditions, this platform cannot perform as efficiently as the GOAC.

**Puromycin aminonucleoside promotes albumin leakage in the GOAC.** To test the hypothesis that our GOAC can model a kidney injury state, we exposed the GOAC to puromycin aminonucleoside (PAN; Fig. 4a), a nephrotoxic agent that alters podocyte morphology and function[25] and can induce focal segmental glomerulosclerosis (FSGS) in mice[46]. When added to GOAC, PAN induced podocyte injury as documented by cytoskeleton rearrangement and loss of permselectivity for albumin at

60 min after stimuli (Fig. 4b–j). The levels of albumin leakage were similar across GOAC with hAKPC-P + hGEC, hiPOD + hGEC, or hpPOD + hGEC (Fig. 4k). Together, these results show that the human GOAC developed in this study mimics function and injury manifestations of the kidney GFB.

**Sera from individuals with MN shows albumin leakage in the GOAC.** To further characterize our system and determine the capacity of GOAC to react to human samples, we tested the response of GOAC to sera from individuals affected by MN, a major cause of nephrotic syndrome (proteinuria with associated

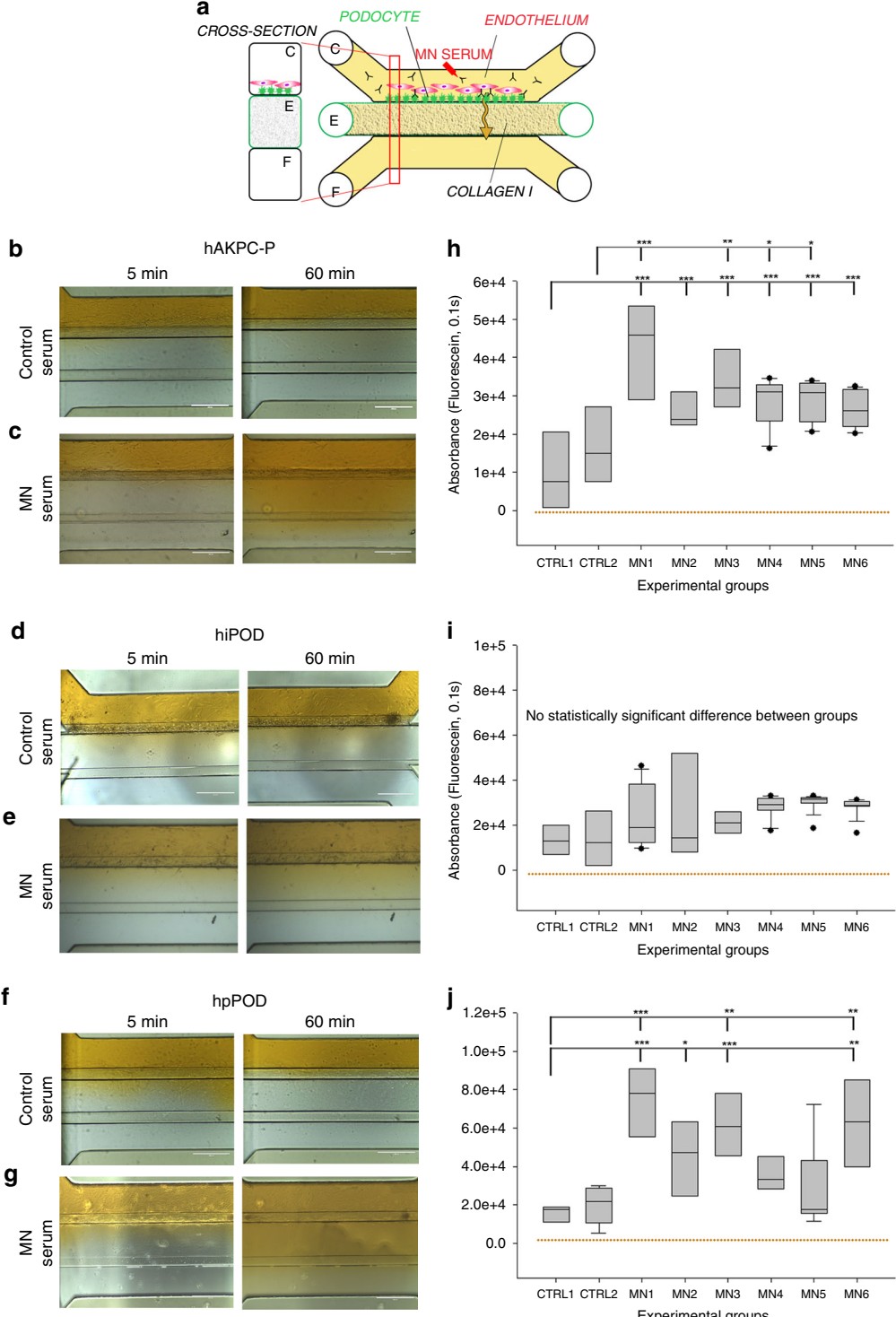

peripheral edema and lipid abnormalities, among other abnormalities) in adults[47]. MN is initiated by the deposition of circulating anti-podocyte autoantibodies in the subepithelial space of the GFB, inducing complement-mediated podocyte injury and proteinuria[48].

First, we tested the ability of IgG to cross the glomerular endothelial cell monolayer on which hGEC were allowed to grow to confluency and form a continuous layer. We confirmed that FITC-IgG added to the top chamber could be detected on the lower chamber after approximately 6 h, thus confirming the

ability of hGEC to allow passage of IgG (Supplementary Fig. 7a), a process also confirmed in vivo[49,50].

After 24 h of exposure to medium containing 0.5% of serum from MN patients or healthy controls (Fig. 5a), the chips with MN serum, but not control sera, showed total IgG (Supplementary Fig. 7b–g) and IgG4 (Supplementary Fig. 7h–m) deposition on the podocytes, recapitulating the main features of MN nephropathy[51]. While chips generated from all podocyte types showed an increase in albumin leakage (Fig. 5b–j), only hAKPC-P (Fig. 5h) and hpPOD (Fig. 5i) chips confirmed a statistically

**Fig. 5** Evaluation of permselectivity using human membranous nephropathy serum samples. **a** Representation of the GOAC albumin permselectivity assay to MN serum exposure. Following a 24 h incubation with media supplemented with 0.5% serum from healthy individuals (CTRL1 and CTRL2) or MN patients (MN1–6), albumin-FITC is applied to channel C (yellow) and flow-through collected in channel F. **b–g** Bright field showing albumin leakage after 5 min (left columns) and 60 min (right columns) after exposure to healthy and MN patients serum in hAKPC-P + hGEC (**b**, **c**), hiPOD + hGEC (**d**, **e**), and hpPOD + hGEC (**f**, **g**) chips. Leakage is evident in hAKPC-P + hGEC and hpPOD + hGEC but not in hiPOD + hGEC chips after exposure to MN serum. **h–j** Box plot graph of fluorescein absorbance (expressed as log) in filtrate collected after 60 min after serum exposure in hAKPC-P + hGEC (**h**), hiPOD + hGEC (**i**), and hpPOD + hGEC (**j**) chips. Statistically significant increase in albumin permeability is evident after exposure to MN sera only in hAKPC-P + hGEC and hpPOD + hGEC chips. Number of replicates for chips used in **h** is as follows: hAKPC-P + hGEC chip and CTRL1: #7; hAKPC-P + hGEC chip and CTRL2: #8; hAKPC-P + hGEC chip and MN1: #4 ; hAKPC-P + hGEC chip and MN2: #7; hAKPC-P + hGEC chip and MN3: #3. hAKPC-P + hGEC chip and MN4: #4; hAKPC-P + hGEC chip and MN5: #7; hAKPC-P + hGEC chip and MN6: #3. Number of replicates for chips used in **i** as follow: hiPOD + hGEC chip and CTRL1: #6; hiPOD + hGEC chip and CTRL2: #4; hiPOD + hGEC chip and MN1: #11; hiPOD + hGEC chip and MN2: #5; hiPOD + hGEC chip and MN3: #4; hiPOD + hGEC chip and MN4: #11; hiPOD + hGEC chip and MN5: #5; hiPOD + hGEC chip and MN6: #4. Number of replicates for chips used in **j** is as follows: hpPOD + hGEC chip and CTRL1: #8; hpPOD + hGEC chip and CTRL2: #9; hpPOD + hGEC chip and MN1: #6; hpPOD + hGEC chip and MN2: #7; hpPOD + hGEC chip and MN3: #7; hpPOD + hGEC chip and MN4: #6; hpPOD + hGEC chip and MN5: #7; hpPOD + hGEC chip and MN6: #7. For **d**, **g**, **j** significant differences were determined by a one-way ANOVA and Holm–Sidak post hoc test, $*p < 0.05$, $**p < 0.01$, $***p < 0.001$ Box plots show the median, the 25th and 75th percentiles, whiskers (median ± 1.5 times interquartile range), and outliers (solid circle)

significant loss of permselectivity following exposure to MN serum while hiPOD (Fig. 5j) failed to respond properly.

We next measured the relationship between the extent of albumin leakage (proteinuria) in our chips (measured as FITC-absorbance in the filtrate collected in channel F) with proteinuria measured in the same patients (Fig. 6) and anti-PLA$_2$R (Phospholipase A2 receptor) titer (Supplementary Fig. 7n–p) by linear regression analysis. We found that proteinuria highly correlated with the diagnostic results obtained in our chips generated using hAKPC-P ($R = 0.8901$, $p < 0.01$, Fig. 6b) and primary podocytes ($R = 0.7995$, $p < 0.05$, Fig. 6d) with a confidence of at least 95%. This correlation was not statistically significant in the chip generated with hiPOD ($R = 0.3155$, $p = $ n.s., Fig. 6c). The same results were obtained when performing the same analysis with anti-PLA$_2$R antibody titer from MN patients, confirming high correlation for hAKPC-P and hpPOD chips but not hiPOD chips (Supplementary Fig. 7n–p).

**Modeling mechanism of MN injury in podocyte in the GOAC.** We then tested whether the GOAC could be used to perform cell-based investigations to delineate mechanisms responsible for podocyte damage and disruption of the filtration barrier. PLA$_2$R is the major podocyte target antigen in MN patients[52,53] along with less common ones like, for example, THSD7A and NEP1 (refs. [52,53]). We confirmed that hAKPC-P, hiPOD, and hpPOD express PLA$_2$R both before and after seeding on the chip (Supplementary Fig. 7q–v). However, when western blotting analysis was performed to quantify PLA$_2$R expression, we found that hiPOD exhibit the lowest expression of PLA$_2$R (significantly different when compared to hpPOD or hAKPC-P, $p < 0.05$). A lower expression of antigen by hiPOD (Supplementary Fig. 7w, x) might explain their limited response to MN.

In MN, following autoantibody binding, several mechanisms are triggered in podocytes, like the complement signaling[54] that can lead to delocalization of nephrin with loss of the slit diaphragm structure and podocyte injury[55]. Indeed, following exposure to MN serum, we confirmed an increase in the expression of C3d protein in the three podocyte lines that, although not significant, suggests an activation of the complement pathway consistent with the in vivo cascade signaling[54,56] (Fig. 6e, f). Complement activation was paralleled by a decrease in nephrin compared to cells cultured with control (healthy) serum (Fig. 6g, h, Supplementary Fig. 8a–f). To further characterize cellular response to MN serum we evaluated SNAIL expression and confirmed its increase in the nuclear region following MN exposure for 24 h (Supplementary Fig. 8g–l). Interestingly, increase in SNAIL appears to be much more marked in

hAKPC-P compared to hiPOD, and could possibly explain the lower proteinuria levels detected in the hiPOD chips (Fig. 5g, Supplementary Fig. 8i, j). These data, taken together, suggest that within the chip it is possible to explore disease mechanisms and cascade signaling. Nonetheless, to fully characterize the disease mechanism and cascade signaling in MN-mediated injury, further investigation is needed (including inclusion of different time points and pathways) and goes beyond the scope of the current work.

While the podocyte is the initial target of autoantibodies in MN, endothelial cells in the glomeruli of affected patients also show signs of injury[57]. To test whether the same occurs in GOAC exposed to MN serum, we measured WGA expression in endothelial cells and found that the expression declined at 24 h after exposure, a phenomenon that did not occur in the presence of control sera (Supplementary Fig. 9). Altogether, these data validate our system as a feasible model to study MN pathophysiology in vitro.

**GOAC response specificity to sera from various CKD.** To exclude that albumin leakage induced in GOAC by MN serum was due to an unselective response to serum from subjects affected by CKD, we exposed the chip to sera from individuals with AS, polycystic kidney disease (PKD), or FSGS. Both AS and PKD are due to a primary renal defect and circulating factors in the serum are not thought to play a role in disease pathogenesis. While putative circulating factors have been described in FSGS, the sera included in the experiments were obtained from individuals with diseases remission; therefore, these factors (if present) were not enough to induce proteinuria in vivo. As shown in Fig. 7a, b, we found that serum from FSGS, AS, or PKD subjects did not trigger loss of permselectivity in our experimental settings, further confirming specificity of albumin leakage induced by MN sera.

**Modeling response to therapy in the GOAC exposed to MN sera.** Building upon the previous results, we tested the hypothesis that GOAC represents a unique platform for screening drugs targeting the GFB. We exposed the GOAC to MN serum for 24 h in the presence or absence of α-melanocortin stimulating hormone (α-MSH). This hormone mimics the activity of the adrenocorticoid hormone (ACTH), clinically used in MN patients to reduce proteinuria[58]. α-MSH main mechanism of action is through inhibition of RhoA inhibitor p190RhoGAP activity, which is crucial for the stabilization of podocyte stress fibers[59]. We found that, similarly to data obtained in vivo using ACTH[60],

| Sample | Sex | Age | Proteinuria (g/24 h) | Creatinine (mg/dl) |
|--------|-----|-----|----------------------|--------------------|
| CTRL1 | M | 43 | 0 | 1.2 |
| CTRL2 | F | 32 | 0 | 0.8 |
| MN1 | M | 73 | 13.4 | 0.9 |
| MN2 | F | 30 | 8.6 | 0.67 |
| MN3 | M | 70 | 6.51 | 0.97 |
| MN4 | M | 47 | 4 | 0.77 |
| MN5 | F | 41 | 2.4 | 0.76 |
| MN6 | F | 30 | 3.6 | 0.53 |

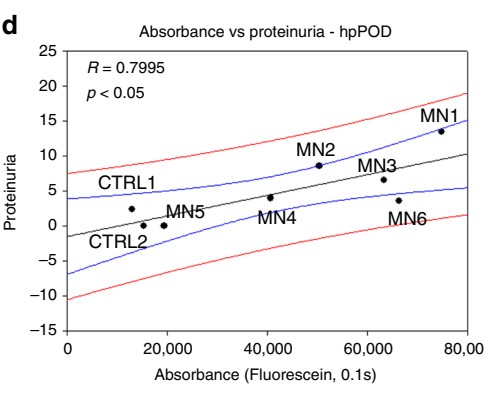

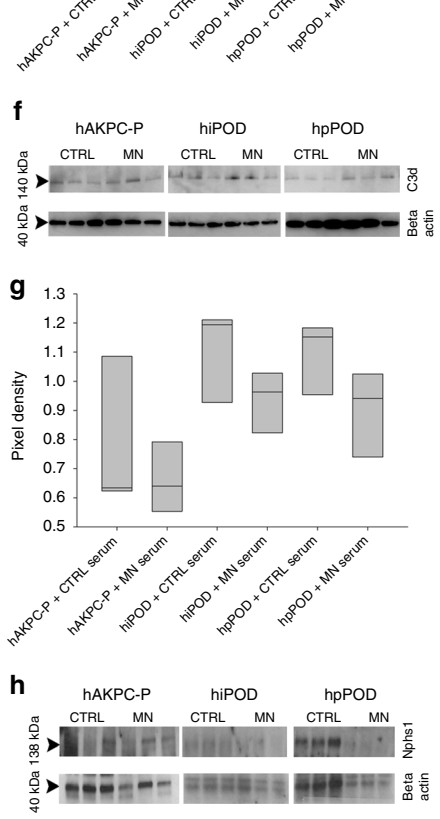

α-MSH prevented proteinuria elicited by MN serum (Fig. 7c, d) indicating that it acts downstream the formation of immune complexes on podocytes. These data demonstrate that GOAC responds to human nephrotoxic serum and nephroprotective treatment similarly to the in vivo human glomeruli.

**Establishing a model of diabetic nephropathy in the GOAC.** Hyperglycemia is recognized as a key initiation factor of ESRD in diabetic patients, which contributes to the increased albumin leakage across the GFB[61]. High glucose has been shown in vitro to

lead to podocyte damage[62,63]. To test whether our system could replicate glucose-induced damage, we exposed GOAC to medium containing 10, 15, or 20 mM glucose and assessed its effect on permselectivity. After 72 h, the chips presented with a significant loss of albumin permselectivity compared to the control group (10 mM) or 15 mM (Fig. 8a, b).

**Primary podocyte mutations promote albumin leakage in GOAC.** To test disease-modeling applications, we generated a GOAC using hAKPC-P derived from a patient affected by AS

**Fig. 6** Correlation of GOAC proteinuria with clinical data and MN mechanism modeling on the chip. **a** Table of clinical parameters for proteinuria in MN serum samples used on the GOAC. **b** hAKPC-P + hGEC chip proteinuria for CTRL1-2 and MN1-6 clinical proteinuria levels suggests a very strong correlation between clinical profile and response in the chip (measured as albumin leakage). R: 0.8901, P < 0.01. **c** hiPOD + hGEC chip proteinuria for CTRL1–2 and MN1–6 clinical proteinuria levels suggests a weak correlation between clinical profile and response in the chip (measured as albumin leakage). R: 0.3156, not significant. **d** hpPOD + hGEC chip proteinuria for CTRL1–2 and MN1–6 clinical proteinuria levels suggests a strong correlation between clinical profile and response in the chip (measured as albumin leakage). R: 0.7995, p < 0.05. For all samples, regression analysis was performed. Equation: Polynomial, linear. Blue lines = 95% confidence band; red lines = 95% prediction band. **e, f** Western blot analysis for C3d (140kDA) and beta actin (40 kDa) in hAKPC-P + hGEC, hiPOD + hGEC, and hpPOD + hGEC chips exposed to healthy or MN serum confirmed increased expression for C3d by all three MN chips. Number of replicates per experimental group: 3 (**f**). Quantification of C3d expression was performed by measuring pixel density and followed by normalization against beta actin. **g, h** Western blot analysis for NPHS1 (138kDA) and beta actin (40 kDa) in hAKPC-P + hGEC, hiPOD + hGEC, and hpPOD + hGEC chips exposed to healthy or MN serum confirmed decreased expression for NPHS1 by all three MN chips. Number of replicates per experimental group: 3. **h** Quantification of NPHS1 expression was performed by measuring pixel density and followed by normalization against beta actin (**g**). For all samples lack of significant differences was determined by a one-way ANOVA and Student–Newman–Keuls post hoc test. Box plots show the median, the 25th and 75th percentiles, whiskers (median ± 1.5 times interquartile range), and outliers (solid circle). Western blot images were cropped to show the relevant bands and improve clarity. Uncropped and unprocessed scans for western blotting analysis are provided in Supplementary Figs. 13, 14

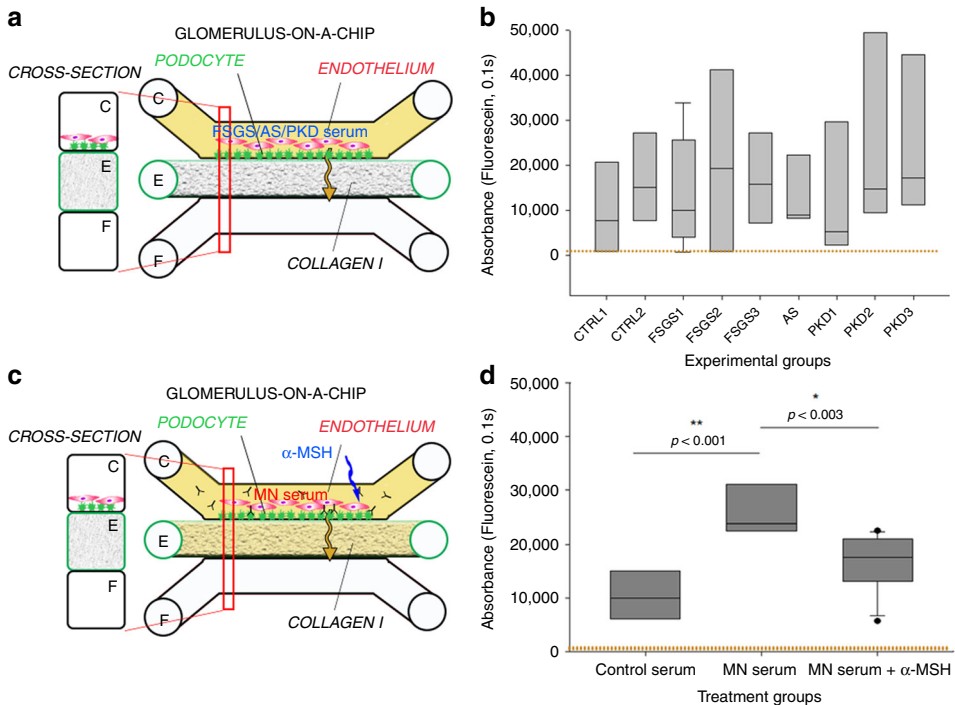

**Fig. 7** Validation of the hAKPC-P GOAC system as a diagnostic and drug screening platform. **a** Scheme of GOAC albumin permselectivity assay and exposure to serum from patients affected by FSGS, AS, and PKD. Following a 24 h incubation with media supplemented with 0.5% serum from healthy individuals (CTRL1 and CTRL2) or CKD patients, albumin-FITC is applied to channel C (yellow) and flow-through presents in channel F. **b** Box plot graph of fluorescein absorbance in filtrate after 60 min following 24 h incubation with serum from healthy individuals (CTRL1 and CTRL2), patients affected by FSGS (FSGS1, FSGS2, and FSGS3), AS, and PKD (PKD1, PKD2, and PKD3). As expected, no statistically significant differences were detected among groups. Number of replicates for chips used in **b** as follow: CTRL1: #7; CTRL2: #8; FSGS1: #9; FSGS2: #3; FSGS3: #4; AS: #4; PKD1: #5; PKD2: #5; PKD3: #5. **c** Scheme of GOAC albumin permselectivity assay and exposure to healthy or MN serum with or without α-MSH. Following a 24 h incubation with media supplemented with 0.5% serum from healthy individuals (CTRL2), MN patient (MN3) or MN patient (MN3) + α-MSH, albumin-FITC is applied to channel C (yellow) and flow-through presents in channel F. **d** Box plot graph of fluorescein absorbance in filtrate after 60 min following 24 h incubation with serum from healthy individuals or MN with or without supplementation with 10 ng/ml of α-MSH for 24 h. Quantification of proteinuria (albumin-FITC) was performed by measuring absorbance (fluorescein, 0.1 s) of flow-through for hAKPC-P + hGEC at 60 min Number of replicates for chips used in **d** is as follows: CTRL2: #6 : MN3: #9; MN3 + a-MSH: #11. Significant differences were determined by a one-way ANOVA and Holm–Sidak post hoc test. Box plots show the median, the 25th and 75th percentiles, whiskers (median ± 1.5 times interquartile range), and outliers (solid circle)

(AS-hAKPC-P). In AS, a mutation on COL4α3α4α5 genes leads to deposition of a defective GBM, leading to CKD and ESRD[26]. As predicted, chips generated using AS-hAKPC-P exhibited a marked and statistically significant albumin leakage (Fig. 8c, d), while chips with podocytes from control individuals did not. Overall, these data document that GOAC may be used to model in vitro abnormalities in the GFB due to genetic abnormalities in podocytes.

## Discussion

Our data show that we can combine human podocytes and glomerular endothelial cells with the MIMETAS™ technology to create a functional GFB in vitro. These cells co-cultured in our GOAC maintain their phenotype and function for at least a month, allowing for long-term experiments. Compared to other proposed glomerular chips[20–22], our system presents important

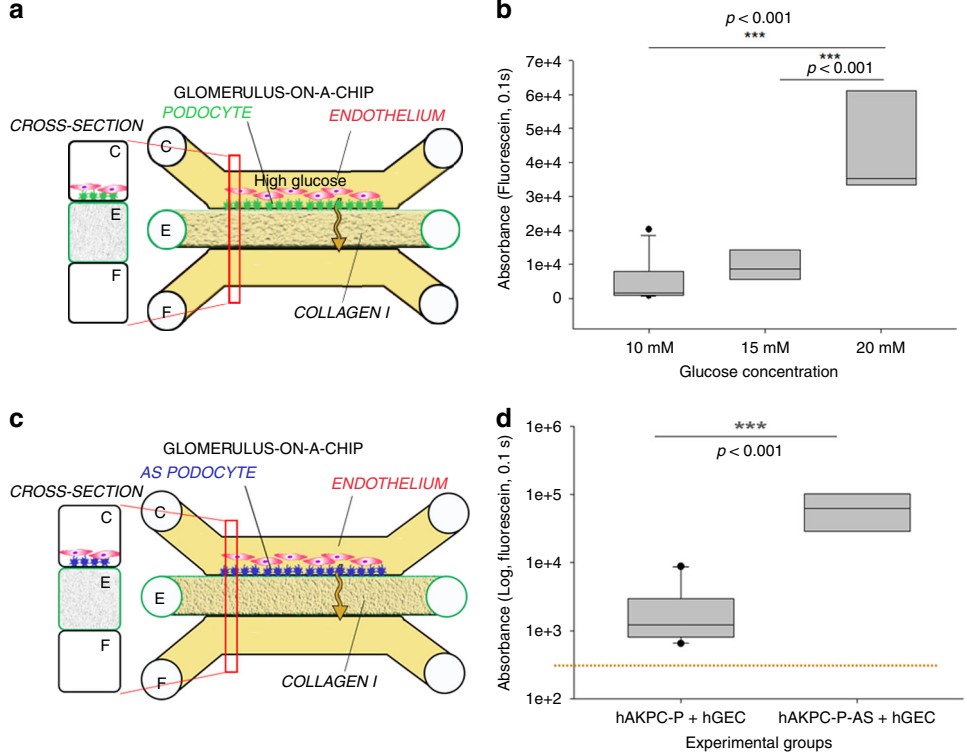

**Fig. 8** Validation of the hAKPC-P GOAC system as disease-modeling platform (221). **a** Scheme of GOAC albumin permselectivity assay and exposure to glucose at different concentrations (10, 15, 20 mM). Following a 72 h incubation, albumin-FITC is applied to channel C (yellow) and flow-through presents in channel F. **b** Box plot graph of fluorescein absorbance in filtrate collected after 60 min following 72 h incubation with 10–20 mM glucose. Number of replicates for chips used in **b** is as follows: 10 mM: #7; 15 mM: #8; 20 mM: #9. **c** Scheme of GOAC albumin permselectivity assay with AS-hAKPC-P + hGEC. Following formation of the AS podocyte-endothelial cell layer, albumin-FITC is applied to channel C (yellow) and flow-through presents in channel F. **d** Box plot graph of fluorescein absorbance in filtrate after 60 min. Quantification of proteinuria (albumin-FITC) was performed by measuring absorbance (fluorescein, 0.1 s) of flow-through for AS-hAKPC-P + hGEC at 60 min Number of replicates for chips used in **d** is as follows: hAKPC-P = hGEC: #12: AS-hAKPC-P + hGEC: #7. Significant differences were determined by a one-way ANOVA and Holm–Sidak post hoc test. Box plots show the median, the 25th and 75th percentiles, whiskers (median ± 1.5 times interquartile range), and outliers (solid circle)

differences and great advantages. The use of a chip devoid of any artificial membrane that separates layers of cells facilitates the correct interactions and crosstalk between cells as it is occurring in vivo. Seeded podocytes form slit diaphragm and endothelial cells form capillary-like structures, cellular features that are essential for a correct filtration activity of the glomerular barrier. We also show the formation of a correctly assembled basement membrane containing specific glomerular extracellular proteins (like COL4α3α4α5 and LAM5α2β1γ). We demonstrated the resemblance of the GFB structure in vitro, thus showing that the absence of synthetic membrane could be essential for the correct assembly of the GFB.

Importantly, we demonstrated the filtration of inulin and the retention of albumin within the GOAC, thus resembling the human GFB. This successful reconstruction of the GFB by our chips can be explained not only by the absence of a synthetic membrane between the cell layers and by the correct assembly of the GBM, but also by the choice of the podocyte source. hAKPC-P present phenotype and function very similar to primary podocytes, and most importantly they can be efficiently differentiated in large scale without the use of any immortalization or laborious protocols, as we have previously shown[25]. We also confirmed our system to be superior compared to transwell systems built following the same protocol, thus further highlighting the strength of our platform. Our data also indicate that immortalized podocytes present some limitations

in studies of permselectivity in response to damage as shown in Figs. 5 and 6.

Of major relevance for research on glomerular diseases, we showed that the GOAC responded to chemical injury with PAN, glucose-induced damage, and nephrotoxic serum from MN patients similarly to in vivo glomeruli. Upon deposition of IgG on the podocytes, the GFB exposed to sera from MN lost permselectivity to albumin to an extent that was proportional to the levels of anti-PLA$_2$R IgG and severity of proteinuria measured in vivo in the same patients. Altogether, these data demonstrate that this system represents a unique platform to study the pathophysiology of glomerular diseases in a manner that, differently from previously proposed works, allows to study (1) changes in 3D conformation of podocytes, endothelial cells, and GBM; (2) abnormalities in their function; and (3) the crosstalk among them. The ability to separately manipulate the three elements of the GFB in the GOAC offers the opportunity to test mechanisms of glomerular diseases by selectively targeting—on a large scale—genes or molecules in the cultured podocytes and/or endothelial cells. The strong association between functional data in the GOAC and in vivo renal parameters also indicates that this system may be used as a platform to identify new biomarkers of glomerular injury in response to various stimuli and to test glomerular toxicity of new compounds.

The fact that a podocyte-targeting drug like α-MSH, clinically used to reduce proteinuria in MN patients, prevented the

proteinuric effects of MN sera in the GOAC is indeed remarkable. This result supports the use of our GOAC for drug screening studies, a major unmet need for research in nephrology. Our system, devoid of synthetic membrane but with functional GBM, allows real doses of drugs to be tested; they can be added directly into channel C without worries that synthetic membrane (with pore size bigger than what is present in the in vivo GBM and lack of the morphology represented by the curvature of the vasculature[22]) could improperly absorb/retain different molecules, compromising evaluation of efficacy and toxicity of the tested compound. Moreover, while organoids have proven to be a key tool for understanding kidney development and for disease modeling, their usefulness for drug screening is still unclear due to the (1) potential incomplete differentiation of the renal structures, (2) difficult diffusion of drugs within the 3D tissue as well as the (3) difficult assessment of proteinuria in their system[64].

Our system also offers a unique prospect for disease-modifying studies. Here, we validate the GOAC using podocytes derived from a patient affected of AS (ref. [26], male X-linked, missense mutation, p.Gly370Glu). We demonstrated that this Alport-GOAC shows improper filtration; thus, it can serve as a platform for studies of personalized medicine. Amniotic fluid can be collected from Alport patients carrying the natural mutation, thus accurately recapitulating the human disease. Most importantly, the derivation of multiple cell lines from patients affected by different mutations of the same disease guarantees representation of disease heterogeneity in real time in a dynamic system, which is not possible for example in mice studies. Even if iPs technology presents this advantage too, the derivation of podocytes from iPs requires immortalization and laborious induction protocols, while hAKPC-P differentiation is an easy and fast process with minimal cell manipulation[25].

We acknowledge that our system has some limitations, including the bi-directionality of the flow (in vivo the flow in the vascular lumen is uni-directional with the glomerulus receiving its blood supply from afferent arterioles before exiting into efferent arterioles and is not recirculated), and the fact that GOAC does not include mesangial cells, an important component of normal glomeruli[65]; future studies will focus on generating a four lane chips, allowing culture of three different cell types. MIMETAS has already demonstrated the feasibility of producing functional proximal tubules[19,66] using their technology so the next step will be to combine the GOAC and the tubules to generate a functional nephron on a chip where filtration and reabsorption can be evaluated at the same time. Of note, to establish the model and generate all the data for the current manuscript we have used about 50 plates, for a total of 2000 independent chips analyzed. Thus, we believe that the high number of chips analyzed allows us to interpret our data with high reproducibility. In conclusion, our chip represents a transformative system that mimics the human renal filtration barrier and is an ideal tool to study glomerular disease mechanisms and drug screening. Chips generated with diseased podocyte lines will increase our understanding of the cellular and molecular mechanisms responsible for glomerular injury and podocyte loss and will advance the design and evaluation of therapeutics strategically targeted to the glomerulus, thus ultimately benefiting patients affected by CKD and renal failure.

## Methods

**Ethics statement and acquisition of human samples**. *Amniotic fluid-derived cells*: Discarded samples of human amniotic fluid from male fetuses (15–20 weeks of gestation) were provided to our laboratory by Labcorp (now Integrated Genomics, Monrovia, CA, USA) after karyotyping analysis. The study was approved by the Children's Hospital Los Angeles (CHLA) Institutional Review Boards and exemption was obtained since no written or verbal consent was required as samples were de-identified. Samples presented with normal karyotype and ultrasound and

were confirmed negative for infectious diseases. Samples of amniotic fluid from patients affected by AS were obtained through the Telethon Biobank (Siena, Italy) directed by Dr. Renieri and Alport hAKPC-P were derived as described below.

*Primary glomerular cells*: Kidneys deemed non-suitable for transplantation were used for isolation of human primary podocytes and glomerular endothelial cells. CHLA Institutional Review Boards approved tissue collection. Discarded kidneys were harvested from infant patients with a non-nephrological cause of death, and thus our isolation of primary podocytes and glomerular endothelial cells rendered functional cell types.

*Immortalized podocyte lines*: were kindly donated by Dr. J. Reiser (Rush University Medical Center, Chicago, IL).

*Patient serum*: De-identified sera from healthy subjects and from individuals with MN ($n = 6$), FSGS ($n = 3$), PKD ($n = 3$), and AS ($n = 1$) were obtained from Drs. Joaquin Manrique (Biobank Navarrabiomed, integrated in the Spanish National Biobanks Network, Complejo Hospitalario de Navarra, Pamplona, Spain) and Andrea Angeletti (S. Orsola-Malpighi Hospital, University of Bologna, Bologna, Italy). Protocols for the collection of these human samples were approved by the Institutional Review Boards of the two Institutions, and informed consent was obtained from all participants.

**Cells: isolation and culture**. Kidney progenitor cells derived from amniotic fluid (hAKPC) were isolated by co-expression of OB-cadherin, CD24, and podocalyxin[25]. Sorted hAKPC were expanded and differentiated into podocytes (hAKPC-P) by culturing on collagen I (Corning, c#354236)-coated plates in VRADD media: RPMI-1640 (Gibco, c#11875093) supplemented with 5% FBS (Gibco, c#26140079), 1% antibiotic (Gibco, c#15070063), 1,25(OH)2D3 [100 nM, cholecalciferol] (Sigma, c#C9756), all *trans* retinoic acid (ATRA) [1 µM], dexamethasone [100 nM] (Sigma, c#D4902), for up to 30 days. Human immortalized podocytes (hiPOD) were cultured as described by Saleem et al.[28]. Re-differentiation of hiPOD was performed by thermoshifting to 37° for up to 15 days.

Human lung fibroblasts (hFIB) were purchased from LifeLine Cell Technology (#FC-0049) and expanded with Fibrolife Media (LifeLine Cell Technology, c# LL-0001) in tissue culture dishes for up to five passages. Human lung endothelial cells were purchased from ATCC (HuLEC-5a, CRL-3244) and expanded with ATCC basal media (#MCDB131, supplemented with 10 ng/ml Epidermal Growth Factor, 1 µg/ml hydrocortisone,10 mM glutamine, FBS to a final concentration of 10%) in gelatin-coated tissue culture dishes for up to five passages.

Primary podocytes (hpPOD) and glomerular endothelial cells (hGEC) were isolated from discarded human kidney samples through mechanical and chemical digestion. Briefly, the kidneys were minced and digested in 125 U/ml collagenase I (Worthington, LS004197) in RPMI-1640 at 37 °C for 30 min and filtered three times in 100-µm cell strainers and once on the 40-µm cell strainer (Corning, c# 352360, 352340). The glomeruli that remained on the 40-µm cell strainer were washed out with PBS and centrifuged at $1800 \times g$ for 7 min The extracted glomeruli were thoroughly checked by light microscopy to confirm the absence of contaminants including afferent and efferent vessels and tubules. The glomerular pellet was re-suspended and plated onto a 100 cm² tissue culture dish in media comprised of RPMI-1640, 5% FBS, and 0.2% Primocin (Invivogen, c# ant-pm-1), and left to incubate overnight at 37 °C. After 24 h the glomeruli were trypsinized (Trypsin-EDTA; Gibco, c# 25200072) using 0.25% trypsin-EDTA for 5 min to allow all the components of the glomerulus, including the hpPOD and hGEC, to separate. Cells were prepared for sorting as described under FACS and flow cytometry analysis in Methods. Once sorted, the NPHS1-FITC-positive cells (podocytes) were seeded onto collagen in VRADD medium (as described above) and cultured for no more than one passage; the CD31-647 cells (hGEC) were plated onto gelatin (Cell Biologics, c# 6950) in human endothelial cell medium (Cell Biologics, c# H1168;) and cultured for no more than 10 passages.

**FACS and flow cytometry analysis**. Kidney progenitor cells were isolated from human total amniotic fluid cell populations by triple staining with antibodies detecting OB-cadherin-FITC, CD24-APC and podocalyxin-PE. hpPOD and hGEC were isolated from human glomerular cell suspension by staining with respectively NPHS1-FITC and CD31-AF647 antibodies. Briefly, cells were blocked using 1× human IgG (Sigma c# I2511) for 30 min and then stained with the specified antibodies, 1 µg/1 × 10⁶ cells/100 µl IgG solution unless otherwise specified on the datasheet, for 1 h on ice. Cells were then washed twice in PBS and filtered immediately before sorting. Cells were sorted using a FACSAria sorter. Unstained and single positive controls were used to perform area scaling, exclude auto-fluorescence, and perform fluorochrome compensation when needed. Cells were first gated based on forward and side scatters (FSC/SSC) to exclude dead cells and then gated for FSC-W/FSC-H and SSC-W/SSC-H to exclude potential duplets. Sorting gates were established based on the unstained population for each sample (Supplementary Fig. 10). For flow cytometry analysis, cells were fixed in 4% paraformaldehyde (Santa Cruz Biotechnology c# sc-281692) for 10 min and permeabilized with 0.05% saponin for nuclear proteins (WT1). Cells were then blocked in 1× human IgG solution for 10 min and incubated with either antibody for WT1, nephrin, CD31, EHD3, syndecan-1, and syndecan-4. Analysis was performed on a FACScanto machine using FACSDiva software. Gating strategy was performed as described above. Histogram plots were obtained using FlowJO software.

**Microfluidic chip and cell seeding**. OrganoPlate[TM] culture was performed using three-lane chip with 400 μm × 220 μm channels (Mimetas BV, the Netherlands). Phaseguide[TM] had dimensions of 100 μm × 55 μm. Gel and perfusion channels have a length of 9 and 13 mm, respectively. In all, 1.67 μl of gel composed of 4 mg/ml Collagen I (AMSbio Cultrex 3D Collagen I Rat Tail, 5 mg/ml, c#3447-020-01), 100 mM HEPES (Life Technologies, c#15630-122), and 3.7 mg/ml NaHCO3 (Sigma, c# S5761) was dispensed in the gel inlet (middle) and incubated 20–30 min at 37 °C. hAKPC-P, hiPOD, hpPOD, hGEC, hFIB, and HuLEC were trypsinized using 0.05% trypsin-EDTA (Gibco,c#LS25300062) aliquoted and pelleted (5 min, 1500 × g). The cells were applied to the system by seeding 2 μl of 1.5 × 10[7] of cells/ml in the inlet of the top medium channel. Subsequently, the OrganoPlate[TM] was placed on its side at an angle for 30 min at 37 °C to allow the cells to sediment against the collagen I. This was followed by addition of 50 μl of podocyte differentiation medium to both the inlet and outlet of the top medium channel and the OrganoPlate[TM] was again incubated on its side overnight at 37 °C to complete cell attachment. The following day, hGEC were applied to the system using the same procedure as described above with addition of endothelial cell medium. This created the polarity of the GFB, with endothelial cells oriented toward the vascular channel represented by our plate, and podocytes oriented toward the urinary channel, which had by then layered on top of the collagen. Media described above was changed every 2–3 days such that endothelial cell medium was added to the top inlet and outlet, and podocyte differentiation medium was added to the bottom inlet and outlet, thereby reaching their respective cell types. The OrganoPlate[TM] was placed horizontally in the incubator (37 °C, 5% CO$_2$) on an interval rocker switching inclination every 10 min, allowing bi-directional flow. Medium (50 μl each on inlet and outlet) was refreshed every 2–3 days.

**Assessment of shear stress**. GOAC platform uses a gravity-based perfusion system with a dynamic flow due to periodic 7° tilting. When the plates are levelled to 0° and both volumes are equal no pressure difference exist between the two wells; however, by periodically tilting the plates, a height difference is imposed between liquid levels in connecting walls which results in a pressure difference that causes associated shear stress. This induced shear stress in the microfluidic channels of the OrganoPlate can be estimated using a numerical model proposed and verified by Vormann M.K. et al.[40]. Using this numerical model, we calculated the induced pressure difference between the two volumes of fluid present in the inlet and outlet wells. We calculated the pressure caused by the gravitational pull on a volume of fluid by $P = pgh$ ($p$ = fluid density, $g$ = gravitational constant, and $h$ = height). The flow rate was calculated by $Q = \Delta PRh^{-1}$ ($\Delta P$ = pressure difference and Rh = resistance). The resistance was calculated by $Rh = 12uL(wh^3 (1 - 0.630hw^1))^{-1}$ ($w$ = width, $u$ = fluid viscosity (0.001 kgm$^{-1}$ s$^{-1}$)), $L$ = channel length. Finally, the shear stress $\tau$ (Pa) was calculated by $\tau = 6uQ(wh^2)^{-1}$. The final value of induced shear stress is equal to 0.0117 Pa.

**Immunofluorescence and confocal imaging**. Immunofluorescent staining was performed on OrganoPlate[TM] and chamber slides of representative cell types: following fixation by 4% paraformaldehyde (Santa Cruz Biotechnology c# sc-281692) and serial washes with PBS. Chips/wells of interest were prepared for staining by blocking with 5% bovine serum albumin (Jackson ImmunoResearch Lab, c#001-000-162) in PBS for 30 min Primary, secondary, and pre-conjugated antibodies were diluted in 2.5% BSA Jackson ImmunoResearch Lab. c#001-000-162) as indicated in Table 1. Thirty microliters of solution was added to the top and bottom inlets and outlets of the chips or 100 μl of solution was added directly into the chamber slide wells. Primary antibodies were incubated at RT for 1 h; following serial washes, secondary antibodies were incubated at RT for 30 min. After a final series of washes in PBS, DAPI was applied (1:1000 in PBS; BD Pharmingen, c# 564907) and the OrganoPlate[TM] or the wells were stored at 4 °C until imaged by confocal microscopy (Zeiss 710 microscope) and processed using the ZEN10 software.

**Scanning electron microscopy**. Samples were processed by the University of Southern-California Keck School of Medicine microscopy core. Samples were fixed in half-strength Karnovsky's fixative, post-fixed in 2% OsO$_4$, followed by ethanol dehydration and hexamethyldisilazane drying. Air-dried specimens were mounted on specimen stubs using silver paste and sputter-coated with gold-palladium according to standard procedures. Specimens were visualized by scanning electron microscopy on a JEOL JSM-6390LV instrument (JEOL, MA, USA) operated at 10 kV accelerating voltage.

**Albumin permselectivity assay and inulin permeability assay**. An albumin and inulin permeability assay were established to evaluate the efficiency of the created GFB. The number of chips used for each experiment is described in the corresponding figure legend and summarized in Supplementary Table 1. Media was aspirated from the bottom inlet and outlet, to which PBS was added. Then, media from the top inlet and outlet was aspirated. Fifty microliters albumin-FITC (Millipore Sigma, c#A9771) or inulin-FITC (10 mg/ml, Sigma, c# F3272) was added to the top inlet and outlet, such that the orientation of filtration would be simulated as in native blood flow: from endothelial cells, through podocytes, and into the urinary space of Bowman's capsule. Presence of FITC, and thus albumin or inulin, in the bottom channel indicated a disruption of the GFB. The chips were imaged at 5 and 60 min, during which the plates continued to incubate at 37 °C. At 60 min, media was collected from the bottom inlet and outlet. Absorbance was measured using the Perkin Elmer Victor 3 plate reader using Wallac 1420 workstation software (fluorescein 485/535, 0.1 s). For long-term studies, the same chips were evaluated for permselectivity at respectively 1, 2, 3, and 4 weeks (hAKPC and hpPOD; hiPOD were evaluated for weeks 1 and 2 since after this time frame these chips are not properly functioning) after hGEC seeding. Culture medium was consistently

### Table 1 List of antibodies and assay-specific concentrations

| Antibody | Company | Catalog # | Dilution |
|---|---|---|---|
| CD24 | R&D | FAB5247A | 1:10 |
| OB-cadherin | R&D | FAB17901G | 1:20 |
| Podocalyxin | R&D | FAB1658P | 1:10 |
| CD31 (AlexaFluor-647) | BD Pharmingen | 561654 | IF 1:50 |
| VEGFR2 | Abcam | 2349 | IF 1.5:100 |
| WGA (Rhodamine) | Vector | RL-1022 | IF 1:100 |
| NPHS1 (FITC) | LifeSpan Biosciences | LS-C370063 | IF 1:100 |
| Nephrin (NPHS1) | Invitrogen | PA5-20330 | WB: 1:1000 |
| WT1 | Abcam | ab15249 | IF 1:50 |
| Col IV α1,2 | Abcam | ab6311 | IF 1:50 |
| Col IV α3 | Shigei Research Institute | H31 | WB: 1:100 |
| Col IV α4 | Shigei Medical Research | RH42 | IF: 1:25 |
| LAM α5 | Abbiotec | 251457 | IF 1:100 WB 1:500 |
| PLA$_2$R | Millipore Sigma | MABC942 | IF 1:25 |
| PLA$_2$R | LifeSpan Biosciences | LS-C153547 | WB: 1:500 |
| IgG4 | LifeSpan Biosciences | LS-C351418-500 | IF: 1:20 |
| IgG (FITC) | Abcam | ab97174 | IF 1:20 |
| F-actin | Life Technologies | r37122 | 1 drop/ml |
| B-actin | GeneTex | GTX109639 | WB: 1:1000 |
| C3d | Abcam | ab17453 | WB: 1:1000 |
| Heparan sulfate | Abcam | ab23418 | IF: 1:75 |
| Syndecan-4 | ThermoFisher | 36-3100 | IF: 1:75 FC: 1:100 |
| Syndecan-1 | Abcam | Ab34164 | IF: 1:75 FC: 1:100 |
| EHD3 | Atlas Antibodies | HPA049986 | IF: 1:100 |
| EHD3 | ThermoFisher | PA5-25963 | FC: 1:50 |
| BAX | Santa Cruz Biotechnology | SC-493 | IF: 1:100 |

replaced every 3 days. After each reading, as described above, the albumin-FITC solution and PBS were removed from the top and bottom inlets and outlets and chips were carefully rinsed with PBS twice to remove excess albumin-FITC before returning to fresh culture medium. Efficiency of the GOAC was calculated by assigning a value of 0 to null fluorescein absorbance readings and a value of 100 to fluorescein absorbance readings equal or higher than 300,000 (measurement obtained when FITC-albumin in freely diffusing through cell-devoid chips, Fig. 3h—COL1 + no cells). Efficiency at each time point is expressed as % ± SEM.

**Transwell establishment.** Following coating of the transwells (Costar, #3495) with collagen I, hpPOD were seeded in VRADD media. Once a monolayer was formed (48 h), we added hGEC and allowed them to attach on top of the podocytes for 7 days. VRADD media was substituted with GEC media, as performed on the GOAC. The transwells were then transferred onto the same rocker used to generate the flow in the GOAC. After 7 days, albumin leakage was tested under the same conditions of the chips (timing, BSA-FITC concentration), filtrate was collected after 1 h, and absorbance measured as described above.

**PAN injury.** In all, 10 μg/ml of PAN, a nephrotoxic molecule (Cayman Chemical c# 15509), was supplemented to the media for 5 days. Media without PAN was used as a control. The number of chips used for each experiment is described in the corresponding figure legend and summarized in Supplementary Table 1. After PAN injury, damage was assessed using the albumin assay performed on the chip as described above.

**Assessment of IgG passage through the GEC layer.** To assess ability of IgG to cross a monolayer of hGEC, 50 μl of gel composed of 4 mg/ml Collagen I (AMSbio Cultrex 3D Collagen I Rat Tail, 5 mg/ml, c#3447-020-01), 100 mM HEPES (Life Technologies, c#15630-122), and 3.7 mg/ml NaHCO₃ (Sigma, c# S5761) was dispensed on top of 24-well transwells (Corning, c# 29442-129) and incubated 20–30 min at 37 °C. In total, 100,000 hGEC were seeded for 3 days or until full confluency on the transwells and supplemented with GEC media. In all, 1 mg/ml of human IgG (Sigma c# I2511) or mouse IgG (Thermofisher, c# 31903) were FITC-labeled using Zenon® labeling technology (Thermofisher, c# Z25402 and Z25002). Fifty microliters of labeled IgG were added onto the top of the transwells and were incubated at 37 °C for up to 24 h. Transwells devoid of cells were used as controls. At 15 min, 30 min, 1 h, 3 h, 6 h, and 24 h media was collected from the bottom of the transwell. Absorbance was measured using the Perkin Elmer Victor 3 plate reader using Wallac 1420 workstation software (fluorescein 485/535, 0.1 s) as described above.

**Experiments with human sera.** The number of chips used for each experiment is described in the corresponding figure legend and summarized in Supplementary Table 1. FBS-free endothelial cell medium supplemented with 0.5% human serum from diseased and healthy individuals was added to the top inlet and outlet and was incubated for 24 h. After 24 h, the human serum-supplemented media was removed from the chips and the albumin assay was performed as described above. Healthy patient serum was used as a control.

**Glucose-mediated injury.** Glucose-mediated injury was induced by supplementing glomerular endothelial medium with high-glucose (Sigma, c#5146) at 10 mM (standard RPMI-1640 glucose concentration), 15 mM, and 20 mM. High-glucose media was added to the top inlet and outlet for 72 h. After 72 h, the serum-supplemented media was removed from the chips and the albumin assays was performed as described above.

**α-Melanocyte-stimulating hormone drug rescue.** α-Melanocyte-stimulating hormone (10 ng/mL; Sigma, c#M4135) was added to 0.5% human patient serum-supplemented endothelial medium to rescue the effect of MN serum on the GFB. The chip was incubated for 24 h. After 24 h, the serum-supplemented media was removed from the chips and the albumin assays was performed as described above.

**Western blot analysis.** Total protein from the OrganoPlate™ was collected by adding 125 U/ml collagenase I (Worthington, LS004197) in a radio-immunoprecipitation assay RIPA lysis buffer (Santa Cruz Biotechnology, c#sc-24948) containing a protease inhibitor cocktail (Thermo Scientific, c#78442) and incubated at 37 °C for 30 min. Protein lysates were centrifuged at 17,000 × g, 4 °C for 10 min to obtain the protein suspension. The supernatant was then collected, and total protein concentrated using acetone precipitation. Briefly, four volumes of ice-cold acetone were added to the protein suspension and incubated on ice for 30 min The solution was centrifuged at 13,500 ×g, 4 °C for 10 min, the supernatant discarded, and the pellet was air dried for 20 min The pellet was re-suspended in 100–200 μl of RIPA buffer containing protease inhibitors. Protein extracts were separated on 4–20% pre-cast Protean TGX gels (Bio-Rad, c#456–1094) followed by transfer onto 0.2 μm polyvinylidene fluoride (PVDF) membranes (Bio-rad, c#1704156) using the Trans-blot Turbo transfer system

(Bio-Rad, c#170–4155). Membranes were soaked in methanol 100% for 10 min, quickly rinsed in 0.1% tween 20 (Sigma-aldrich c#P9416), 1× Tris-buffered saline buffer (TBS-T). Blocking was performed in 5% blotto, non-fat dry milk (Santa Cruz Biotechnology, c#sc2325) in TBS-T buffer for 1 h at RT, followed by primary antibody incubation (in 2.5% milk solution) ON at 4 °C in rocking conditions. Following washes in TBS-T buffer (10 min for three times), membranes were blotted with host-specific horseradish peroxidase (HRP)-conjugated secondary antibodies diluted in 2.5% skim milk (in TBS-T) at RT for 30 min. For Col4A3 chain detection, the same electrophoresis and transfer methods were used. The membranes were then processed by blocking with 3% BSA containing 50 mM Tris-HCl buffer (containing 150 mM NaCl) for 30 min Membranes were washed three times with 0.05% tween 20-Tris buffer and blotted ON at 4 °C with COL4A3 antibody diluted in 1% BSA-containing Tris-HCl buffer. The same solution was used to dilute the HRP-conjugated secondary antibody. Signal was detected by using the SuperSignal West Femto substrate (Thermo Scientific, c#34096) and impressed on Amersham Hyperfilm ECL (GE Healthcare, c#28906835). Densitometry was performed on images using ImageJ software. Uncropped and unprocessed scans for western blotting analysis are provided in Supplementary Figs. 11–14.

**Statistics.** Statistical analysis was performed using SigmaPlot v11.2. All graphical data are displayed as the mean + SEM. Normality test (Shapiro-Wilk) and equal variance tests were performed. One-way ANOVA was used to compare independent sets of normally distributed data. Holm–Sidak post hoc test was performed unless otherwise indicated. When a normal distribution was not confirmed, Kruskal–Wallis one-way analysis of variance was performed instead. Studies of correlation across sets of samples were performed by polynomial linear regression analysis. For all statistical analysis, a p value less than 0.05 was considered statistically significant.

**Reporting Summary.** Further information on research design is available in the Nature Research Reporting Summary linked to this article.

## Data Availability
The authors declare that the main data supporting the findings of this study are available within the article and its Supplementary Information files. Extra data are available from the corresponding author upon request.

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

## Acknowledgements

This work is supported by grants from the Alport Syndrome Foundation; TSRI Research Career Development Award; Wright Foundation Pilot Award; and by the GOFARR Foundation and the Schenkman Family. We thank the investigators of the Saban Research Institute, Children's Hospital Los Angeles for the helpful discussion and the staff for ancillary support. We would like to thank J. Marcheque and Dr. Sargis Sedrakyan for their contribution to the experiments. We are grateful to Novabiosis for providing the human kidney samples. We thank the Siena Telethon Network of Genetic Biobank for the Alport Syndrome amniotic fluid samples. We would like to thank Paul Vulto and MIMETAS^TM for technical support. We also thank the CHLA Flow Cytometry Core and the CHLA Imaging core for research support.

## Author Contributions

A.P. performed experiments, contributed to analysis of data and preparation of the manuscript. P.C. provided clinical samples, designed the experiments, contributed to analysis and interpretation of the data, and preparation of the manuscript. V.V. performed experiments, contributed to analysis of data and preparation of the manuscript. A.A. and J.M. provided clinical samples and contributed to the revision of the manuscript. A.R. provided Alport Syndrome cell lines and contributed to the preparation of the manuscript. R.D.F. contributed to analysis and interpretation of the data and preparation of the manuscript. L.P. designed the experiments, contributed to analysis and

interpretation of the data and preparation of the manuscript. S.D.S. designed and performed experiments and contributed to the design, analysis and interpretation of the data and preparation of the manuscript.

## Additional information

**Competing interests:** R.D.F., L.P. and S.D.S. are inventors on a U.S. provisional patent relating to Glomerulus-on-a-Chip, submitted by Children's Hospital Los Angeles. The other authors declare no competing interests.

