## [Peer Review File · Nature Communications]

Reviewers' comments:

Reviewer #1 (Remarks to the Author):

The paper "a glomerulus on a chip to recapitulate the Human glomerular filtration barrier" by Petrsyan et al describes a method in which renal-derived cells, and other cells, are seeded in a microdevice, or chip. The renal cells were obtained by cell sorting and modestly characterized by podocyte specific markers and CD31, specific for amongst other endothelial cells. In essence the two cell types, podocytes and glomerular endothelial cells, that constitute the glomerular filtration barrier are seeded on top of each other in one channel of the chip, with the podocytes as first layer. The podocytes are cultured on a collagen layer present in the underlying channel. Using some podocyte-specific markers and CD31 cells were characterized. Cell injury was induced by PAN, and cells were exposed to serum from 3 patients with membranous nephropathy, and 2 diseases controls, FSGS and Alport. Healthy control serum was used as well. FITC labeled albumin passage was used as a measure for leakage under the different conditions applied. In one case podocytes were derived from a patients with alport syndorome. At first sight the paper looks interesting, but several issues can be raised.

The authors claim that their cells are glomerular endothelial cells. This can be questioned. The method used to isolate glomeruli uses only 2 steps a 100 micrometer sieve and a 40 micrometer sieve. Normally human glomeruli are in the range of 100 micrometers, and normally a series of gradual decreasing sieves are used. There is a serious chance that their glomeruli on sieve 40 actual also contain tubular fragments, but most importantly also vessel part with endothelial cells of non-glomerular origin. The authors can only claim that they have used glomerular endothelial cells when they show the presence of non-diaphragmed fenestrea that are characteristic for glomerular endothelium. The isolation of endothelial cells by CD31 and characterization by CD31 and WGA is very limited does not give evidence that they have used glomerular endothelial cells.

The authors seed endothelial cells on top of podocytes. Normally differentiated podocytes do not form monolayers in vitro, in particular not within the timeframe used by the authors, that is 48 hours. So there should be places where endothelial cells and podocytes are next to each other. In other words, the sandwich contains holes. The authors state that two distinct cell layers can be distinguished. However for me it is impossible to see these layers in any of the pictures provides.

The glycocalyx is hardly addressed (how thick is the layer obtained via WGA staining?)., heparan sulfate, the polysaccharide that gives the barrier function of the glycocalyx is not addressed at all.

The test with serum of Membranous patients suggest that the leakage is independent of anti-PLA2r antibodies, since they were absent in patient 1, whereas this patient gave the largest effect in the leakage assay in Figure 4. When it would be antibody dependent, the endothelial cell layers is most likely also compromised, since the glycocalyx is not facilitating quick diffusion of IgG of 150kDa molecular mass.

Overall the quality of most of the stainings in the manuscript is low, magnifications are low, and difficult to interpret, even not with the help of the cartoon of the chip in which is indicated where what to see.

The 3 patients tested with membranous seems very limited as well as the 2 diseases controls (Alport and FSGS serum), one membranous treatment sample and one chip containing podocytes from one patient with Alport.

The authors state that they have a stable system in culture for weeks, however they do not show these data.

In my opinion the authors severely over-interpret their model as being a bona fide model for the glomerular filtration barrier. The data with patient material is in several cases limited to n=1 observations. The authors do not convincingly show that their systems is superior to other in vitro models (for example trans-wells) mimicking barrier function and allowing albumin permeability measurements and TEER measurements.

Reviewer #2 (Remarks to the Author):

This paper is quite interesting to mimic human glomerular filtration barriers without using any artificial membrane. As far as I know, this structure is a simple and long-term maintaining system. For the success of chip-using experiments, human cell availability is very critical. This paper also shows the relatively easy way to get human glom cells.

They showed some kidney disease models using these glom chips, including chemical injury, antibody-mediated injury and genetic injury. Although a few kidney disease models were shown, still other diseases are challenging to mimic, such as FSGS, and mesangial cell-/immune cell-mediated kidney diseases.

Authors mentioned that this barrier system had been maintained for over 45 days. Were there any changes of barrier functions (perm selectivity) during the whole observation periods?

Short term injury models (1-day exposure of MN sera, 5-day exposure of puromycin) are effective in this system. Long-term injury models may be warranted in the future (ex. Diabetic model).

Can you calculate the shear stress on the top channels? It may affect the formation of Glom barriers and may be very informative to chip researchers.

We would like to thank the Reviewers for their comments, that we have found to be appropriate and helpful to improve the quality of the Manuscript. In this revised version we have addressed their concerns and, specifically, we have 1) performed an additional in-depth characterization of the glomerular endothelial cells to confirm their phenotype, we evaluated the presence of fenestration and performed a characterization of their glycocalyx (Fig S4); 2) performed a thorough assessment of long-term viability, including evaluation of function (Fig. 3i) and apoptosis (S5); 3) established an in vitro system to confirm the ability of IgG to cross the glomerular endothelial cell monolayer (Fig. S7); 4) increased the number of samples for MN experiments from 3 to 6 (Fig. 4-5), PKD from 1 to 3 and FSGS from 2 to 3 (Fig. 6 d) to further support the functionality of our system; 5) set up an additional disease model of glucose-induced damage (Fig. 6e-f) and 6) calculated shear stress within the GOAC.

Please find below our detailed response to each specific comment.

Reviewers' comments:

Reviewer #1 (Remarks to the Author):

The paper “a glomerulus on a chip to recapitulate the Human glomerular filtration barrier” by Petrsyan et al describes a method in which renal-derived cells, and other cells, are seeded in a microdevice, or chip. The renal cells were obtained by cell sorting and modestly characterized by podocyte specific markers and CD31, specific for amongst other endothelial cells. In essence the two cell types, podocytes and glomerular endothelial cells, that constitute the glomerular filtration barrier are seeded on top of each other in one channel of the chip, with the podocytes as first layer. The podocytes are cultured on a collagen layer present in the underlying channel. Using some podocyte-specific markers and CD31 cells were characterized. Cell injury was induced by PAN, and cells were exposed to serum from 3 patients with membranous nephropathy, and 2 diseases controls, FSGS and Alport. Healthy control serum was used as well. FITC labeled albumin passage was used as a measure for leakage under the different conditions applied. In one case podocytes were derived from a patients with alport syndorome. At first sight the paper looks interesting, but several issues can be raised.

1. The authors claim that their cells are glomerular endothelial cells. This can be questioned. The method used to isolate glomeruli uses only 2 steps a 100 micrometer sieve and a 40 micrometer sieve. Normally human glomeruli are in the range of 100 micrometers, and normally a series of gradual decreasing sieves are used. There is a serious chance that their glomeruli on sieve 40 actual also contain tubular fragments, but most importantly also vessel part with endothelial cells of non-glomerular origin. The authors can only claim that they have used glomerular endothelial cells when they show the presence of non-diaphragmed fenestrea that are characteristic for glomerular endothelium. The isolation of endothelial cells by CD31 and characterization by CD31 and WGA is very limited does not give evidence that they have used glomerular endothelial cells.

We thank the Reviewer for the useful comments. We would like to assure the Reviewer that isolation of glomeruli from different species is a very well-established procedure in our laboratory as described in our previous publications (Da Sacco et al, PlosOne, Sedrakyan et al, Scientific Reports, Sedrakyan et al. JASN). The successful isolation of glomeruli using various pore-sized

sieves ranging from 180-53 μm has been reported (van Setten et al. *Kid. Int* 1997). In this revised Manuscript we have added a Suppl. Figure (Fig. S1b-c) in which we demonstrate the purity of our isolation protocol; it is evident that isolated glomeruli do not contain tubules, vessels or other contaminants.

To confirm the glomerular origin of the endothelial cells, we have performed an additional characterization of hGEC as shown in Fig.S4. More specifically, we have confirmed that, following isolation, hGEC express EH Domain Containing 3 Protein (EHD3) (Fig. S4a), a marker that has been shown to be expressed only by glomerular endothelial cells in the adult human kidney (as shown in Fig. S4b, Patrakka et al., *JASN* 2007). About 90% of the isolated and expanded hGEC highly express this marker. We have also confirmed the presence of fenestrations with an average size of approximately 60 nm (by SEM), matching the measurement performed in published studies (Satchell et al, *Kid Int*, 2006; Satchell et al, *Am J Physiol Renal Physiol*, 2009).

We believe that our characterization in this revised Manuscript is more comprehensive and demonstrates that we can isolate in a reproducible manner GEC of human origin.

2. The authors seed endothelial cells on top of podocytes. Normally differentiated podocytes do not form monolayers in vitro, in particular not within the timeframe used by the authors, that is 48 hours. So there should be places where endothelial cells and podocytes are next to each other. In other words, the sandwich contains holes. The authors state that two distinct cell layers can be distinguished. However for me it is impossible to see these layers in any of the pictures provides.

As the Reviewer suggests, formation of monolayers by podocytes (immortalized lines or derived by iPS cell) have been reported by several publications in the field, with time-frames ranging from 2 up to 3 weeks (Qian et al, *Sci Rep*, 2019; Hunt et al, *JASN*, 2005). However, this specific time-frame refers to the time required by the cells to fully differentiate into mature podocytes and formation of a monolayer is often used as a parameter to assess maturation. In all our experiments, as described in the Methods section, both hAKPC-P and iPOD cells were differentiated for 14-21 days to obtain a “mature” podocyte prior seeding into the chip as described in our previous publication (Da Sacco et al, *PlosOne*, 2013), thus requiring a shorter time (48 hours) to re-constitute a monolayer in the chip. hpPOD are isolated directly from adult kidney and therefore are already mature and terminally differentiated podocytes. For example, in the work from Musah et al, iPS are differentiated for 16 days to reach the intermediate mesoderm stage before being transferred within the chip. Podocytes are then induced for additional 5 days within the chip to reach the mature phenotype (Musah et al, *Nat Biomed Eng.* 2017). In summary, the different podocyte lines described in the current Manuscript (hAKPC-P, iPOD and hpPOD) are already “mature podocytes” when seeded on the chip and need shorter time to generate a monolayer. We also believe that a 3D culture system (as our chip) facilitates the formation of the monolayer in a shorter time compared to 2D culture.

3. The glycocalyx is hardly addressed (how thick is the layer obtained via WGA staining?), heparan sulfate, the polysaccharide that gives the barrier function of the glycocalyx is not addressed at all.

We thank the Reviewer for the comment. In glomerular endothelial cells, proteoglycans like Syndecan-1 and Syndecan-4 (Satchell et al., *Nat Rev Neph*, 2013; Singh et al, *JASN*, 2007) along

with Heparan Sulfate (Rops et al., *Kid Int* 2014) are fundamental part to the glycocalyx. The loss of these molecules occurring during kidney disease, leads to damage of the permselectivity (Lipphardt M. et al, *Am J Physiol Heart Circ Physiol*. 2018; Jing et al. *Cell Signal*. 2016; Kuwabara et al. *Diabetologia*, 2010). Thus, considering their fundamental role of these components of the glomerular endothelial glycocalyx, we have further assessed their expression by immunofluorescence and flow cytometry as shown in Fig. S4. We have confirmed by flow cytometry that 99.7% and 96.8% of hGEC are positive for Syndecan-1 and Syndecan-4 respectively. Immunofluorescence staining was used to confirm these results and to assess presence of heparan sulfate on hGEC.

We also added in panel c (Fig. 2) a clearer picture of WGA staining and included a higher magnification. We measured the thickness of the WGA layer and confirmed it to be $\sim 0.5 \mu\text{m}$, compatible with other publications that reported a thickness of 0.2-0.4 μm in immortalized human glomerular endothelial cells (Singh et al, *JASN*, 2007).

4. The test with serum of Membranous patients suggest that the leakage is independent of anti-PLA₂r antibodies, since they were absent in patient 1, whereas this patient gave the largest effect in the leakage assay in Figure 4. When it would be antibody dependent, the endothelial cell layers is most likely also compromised, since the glycocalyx is not facilitating quick diffusion of IgG of 150kDa molecular mass.

Primary MN is caused by the subendothelial deposition of auto-antibodies against podocyte antigens. While PLA₂R is the most known common antigen responsible for the auto-immune response (accounting for $\sim 75\%$ of the MN clinical cases), target antigens include also THSD7A and others not yet identified. This is the case of patient 1 and patient 5 here reported, who did not display autoantibodies against known target antigens. Therefore, while the serum from these patients are not informative on the pathogenic role of anti-PLA₂R antibodies, they still support that concept that anti-podocyte antibodies (verified by total IgG and IgG4 deposition on podocytes in the chip; Figure S7) promote proteinuria.

*In regard to the ability of IgG to cross biological barriers, including the glomerular endothelial barrier, evidence exists that endothelial cells mediate IgG uptake and crossing of integral barriers (Tuma et al., *Physiol Rev*, 2003). It has been shown that while IgG dimers cannot cross the glomerular endothelial barrier and reach the podocyte, IgG monomers can reach the podocytes, to a certain extent (Lawrence et al., *PNAS*, 2017). To elucidate the ability of IgG to cross a glomerular endothelial monolayer, we have established an in vitro system based on the formation of an endothelial monolayer (the same cell line used for the chip experiments) on transwells and added FITC labeled human and mouse (as additional control) IgGs to the top chamber. A transwell system was deliberately chosen for this specific experiment to allow the sampling of the filtrate (50 μl /measurement for each timepoint) at multiple timepoints over the 24 hours, to better inform us about the passage of IgG through the hGEC barrier. We have tracked passage of the IgG at different timepoints to confirm ability of the endothelial cells to allow IgG passage to the bottom chamber. As shown in Fig. S7a, about 6 hours are required to detect FITC-labeled IgG in the bottom chamber, suggesting that passage of IgG is indeed occurring through the monolayer. Transwells coated with collagen I but devoid of cells, as well as transwells with no cells nor collagen I showed a rapid IgG diffusion within 30 minutes. The fact that 6 hours are required for the IgG to cross the endothelial cell layer suggests that passage of is not mediated by simple diffusion, but rather occurs through other mechanisms. While the study of these mechanisms is*

beyond the purpose of the current manuscript, we believe that these newly generated results support our working model.

5. Overall the quality of most of the stainings in the manuscript is low, magnifications are low, and difficult to interpret, even not with the help of the cartoon of the chip in which is indicated where what to see.

We thank the Reviewer for the feedback. To address the issue of low quality of some pictures, we have repeated immunofluorescence staining and replaced some of the pictures in figure 2. Specifically, we have added a new confocal image for WGA staining (Fig. 2d) to improve its clarity and we have included a magnification panel for figures 2j-k-l.

6. The 3 patients tested with membranous seems very limited as well as the 2 diseases controls (Alport and FSGS serum), one membranous treatment sample and one chip containing podocytes from one patient with Alport.

In our original submission we tested serum from 2 control patients, 3 MN patients, 2 FSGS patients, 1 Alport Syndrome patient and 1 PKD patient for a total of 9 distinct serum samples. We also tested podocytes differentiated from healthy and Alport Syndrome individuals' amniotic fluid. The number of chips used for each experiment (as reported in each legend) is between 4 and 22. To clearly indicate the types of the serum and replicates used for each disease, we now have added a table on the Supplementary Materials reporting specific number of chips used for each experimental group. We would like to point out that our chip plate allows 40 chips to be run simultaneously (see figure). To improve the relevance (and reproducibility) of our results, we have included 3 additional MN serum samples (Fig. 4 and Fig.5), 1 more FSGS serum sample and 2 PKD serum samples in Fig. 6.

[REDACTED]

7. The authors state that they have a stable system in culture for weeks, however they do not show these data.

We thank the Reviewer for the useful comment. To address the question, we have included a long-term study of permselectivity. As shown in Fig. 3i, permselectivity can be maintained for at least 4 weeks with limited loss of impermeability in both hAKPC-P and hpPOD GOAC. On the other hand, GOAC using hiPOD showed a more rapid decline of efficiency that rendered them a poorer choice already at 2 weeks. Moreover, we have assessed viability of cells within the GOAC at 28 days by apoptotic marker BAX staining. As shown in Fig. S6k-m, the majority of cells within the chip have been found negative for the marker, suggesting that cells are still viable in long-term culture after 4 weeks, compared to a to the 2-week span reported by both Wang et al and Musah et al.

8. In my opinion the authors severely over-interpret their model as being a bona fide model for the glomerular filtration barrier. The data with patient material is in several cases limited to n=1 observations. The authors do not convincingly show that their systems is superior to other in vitro

models (for example trans-wells) mimicking barrier function and allowing albumin permeability measurements and TEER measurements.

One of the peculiar aspects of the microfluidic system used in the current work is the presence of 40 independent chips on the same plate, each supporting adjacent 3 channels (see picture above). This system allows for 40 independent experiments (or replicates) to be run at the same time in a high-throughput fashion. To establish the model and generate all the data for the current Manuscript we have used about 50 plates, for a total of 2000 independent chips analyzed. This high number of chips analyzed allows us to interpret our data with reproducibility and with statistical differences between our experimental groups.

The number of observations for each patient's serum (as reported for each figure in the respective legend as "replicates") ranges between a minimum of 4 to 11 chips. To improve the clarity of the Manuscript, we have added a table on the Supplementary Materials that reports the number of chips used for each experiment.

To further highlight the advantages of our platform in comparison to other in vitro models and to confirm that our GOAC replicates in bona fide the permselectivity of the glomerular filtration barrier, we have generated podocyte-endothelial barriers on 24-well transwells using the same protocol (including cell isolation, media and timing) used for our chips. Briefly, following coating of the transwell with collagen I, primary podocytes were seeded in VRADD media. Once a monolayer was formed (48 hour), we added glomerular endothelial cells and allowed them to attach on top of the podocytes for 7 days. VRADD media was substituted with GEC media, as performed on the GOAC. The transwells were then transferred onto the same rocker used to generate the flow in the GOAC. After 7 days, albumin leakage was tested under the same conditions of the chips (timing, BSA-FITC concentration), filtrate was collected after 1 hour and absorbance measured. As shown in the graph below, all transwells experienced a significant albumin leakage, thus suggesting that, under the same protocol conditions, transwells cannot perform as our GOAC. Displayed for comparison, orange dotted line represents absorbance measured on the chips.

Reviewer #2 (Remarks to the Author):

1. This paper is quite interesting to mimic human glomerular filtration barriers without using any artificial membrane. As far as I know, this structure is a simple and long-term maintaining system. For the success of chip-using experiments, human cell availability is very critical. This paper also shows the relatively easy way to get human glom cells.

They showed some kidney disease models using these glom chips, including chemical injury, antibody-mediated injury and genetic injury. Although a few kidney disease models were shown, still other diseases are challenging to mimic, such as FSGS, and mesangial cell/immune cell-mediated kidney diseases.

We thank the Reviewer for the comments. We acknowledge that many renal diseases are difficult to mimic in an in vitro system. As the Reviewer pointed out, we have already shown (Fig. 6) that our system can be used to study different kidney injury models, including FSGS. In addition to the disease models already described in our Manuscript, we have set up a high glucose injury model to show versatility of our system. Following exposure to 20mM glucose, we confirmed disruption of the barrier as soon as 72 hours, leading to a statistically significant albumin leakage. These results are consistent with the results obtained on a different glomerulus-on-a-chip systems as reported by Wang et al. New data has been added to Figure 6, Material and Methods have been updated and a new paragraph has been included in the Results section.

We would like to underline that one of the advantages of our microfluid system is its high versatility in terms of design, with the ability to increase the number of parallel channels, for additional cell types (including but not limited to mesangial cells or immune cells), thus allowing us to mimic more complex diseases involving non-GFB cells like mesangial cells or proximal tubular structures.

2. Authors mentioned that this barrier system had been maintained for over 45 days. Were there any changes of barrier functions (perm selectivity) during the whole observation periods?

We thank the Reviewer for the useful comment. To further elucidate the effect of long-term culture on functionality of the GOAC, we have added a long-term study of permselectivity. As shown in Fig. 3i, permselectivity can be maintained for at least 4 weeks with limited loss of impermeability in both hAKPC-P and hpPOD GOAC. We would like to highlight that the GOAC using hiPOD showed a rapid decline of efficiency already at 2 weeks, thus possibly indicating that hiPOD are not efficient as the other lines. Moreover, we have assessed viability of cells within the GOAC at 4 weeks by apoptotic marker BAX staining. As shown in Fig. S6k-m, the majority of cells within the chip are negative for the marker, suggesting that cells are still viable in long-term culture. Compared to existing glomerulus-on-a-chip system, our system has shown viability for at least 4 weeks compared to the 2-week span reported by both Wang et al and Musah et al.

3. Short term injury models (1-day exposure of MN sera, 5-day exposure of puromycin) are effective in this system. Long-term injury models may be warranted in the future (ex. Diabetic model).

We thank the Reviewer for the input. In the revised version of this Manuscript we have added additional data to support that our system is viable for more than a month both in terms of functionality (Fig. 3i) and (Fig. S5.k-m). As the Reviewer as suggested, we believe that the long-term maintenance of phenotype and functionality within our system (at least 4 weeks compared to the 2-week span reported by both Wang et al and Musah et al) fully supports the possibility to set up long-term injury models in the near future.

4. Can you calculate the shear stress on the top channels? It may affect the formation of Glom barriers and may be very informative to chip researchers.

Shear Stress plays a critical role in maintaining glomerular function and any significant variation leads to drastic perturbation of glomerular homeostasis. A Previous studies on glomerulus-on-a-chip systems established in other labs have reported a shear stress ranging from 0.003-0.007 dyn/cm². [Zhou et al, Sci. Rep, 2016; Musah et al., Nat. Biomed Eng, 2017] on the top channel. In comparison, when we calculated the shear stress within our 3-channel system based on a previous work by Vormann et al., AAPS J, 2018, we found that our system allows the generation of a shear stress equal to 0.0117 Pascal (or 0.117 dyn/cm², calculated using the formula: $\tau = 6\mu Q / (wh^2$; more details can be found in the Material and Methods section). In glomerular capillaries, shear stress has been estimated to range from approximately 1 to about 95 dyne/cm² (corresponding to 0.1 to 9.5 Pascal), with mean values in most loops of 5 to 20 dyne/cm² (0.5-2 Pascal, Ballermann et al. Kid Int, 1998). While our shear stress value is below the physiological levels found in vivo, we would like to point out that, compared to other glomerulus-on-a-chip systems, its value is the closes

REVIEWERS' COMMENTS:

Reviewer #1 (Remarks to the Author):

Overall the authors did a great job in improving their manuscript. Most of my comments were addressed satisfactorily.

However, the novelty of their setup is still not prominently stressed by the authors. Therefore, I propose that the authors include the core of their following reply (which convinced me of the novelty) both in abstract and discussion. It is not directly clear from the current manuscript now.

""One of the peculiar aspects of the microfluidic system used in the current work is the presence of 40 independent chips on the same plate, each supporting adjacent 3 channels (see picture above). This system allows for 40 independent experiments (or replicates) to be run at the same time in a high-throughput fashion. To establish the model and generate all the data for the current Manuscript we have used about 50 plates, for a total of 2000 independent chips analyzed. This high number of chips analyzed allows us to interpret our data with reproducibility and with statistical differences between our experimental groups. The number of observations for each patient's serum (as reported for each figure in the respective legend as "replicates") ranges between a minimum of 4 to 11 chips. To improve the clarity of the Manuscript, we have added a table on the Supplementary Materials that reports the number of chips used for each experiment.""

In addition I propose to include the comparison data with the transwell system, showing theirs is superior. Notably, this is the point in the reply following the cited one above. The figure can be placed as supplement.

Reviewer #2 (Remarks to the Author):

Answers were appropriate.

No additional comments.

Reviewers' comments:

Reviewer #1 (Remarks to the Author):

Overall the authors did a great job in improving their manuscript. Most of my comments were addressed satisfactorily.

However, the novelty of their setup is still not prominently stressed by the authors. Therefore, I propose that the authors include the core of their following reply (which convinced me of the novelty) both in abstract and discussion. It is not directly clear from the current manuscript now.

""One of the peculiar aspects of the microfluidic system used in the current work is the presence of 40 independent chips on the same plate, each supporting adjacent 3 channels (see picture above). This system allows for 40 independent experiments (or replicates) to be run at the same time in a high-throughput fashion. To establish the model and generate all the data for the current Manuscript we have used about 50 plates, for a total of 2000 independent chips analyzed. This high number of chips analyzed allows us to interpret our data with reproducibility and with statistical differences between our experimental groups.

The number of observations for each patient's serum (as reported for each figure in the respective legend as "replicates") ranges between a minimum of 4 to 11 chips. To improve the clarity of the Manuscript, we have added a table on the Supplementary Materials that reports the number of chips used for each experiment.""

In addition I propose to include the comparison data with the transwell system, showing theirs is superior. Notably, this is the point in the reply following the cited one above. The figure can be placed as supplement.

We would like to thank the Reviewer for the positive feedback.

Following the useful suggestion, we have included both in the Abstract and in the Discussion a paragraph to further stress the novelty of the proposed platform.

We have also included the transwell system experiment in Supplementary Figure 6e and amended both the Methods and Results sections accordingly.

Reviewer #2 (Remarks to the Author):

Answers were appropriate.

No additional comments.

We would like to thank the Reviewer for the positive feedback.